# In vivo CRISPR screening identifies SAGA complex members as key regulators of hematopoiesis

Archana Shankar [1,2,3,14], Leonid Olender[4,5,14], Ian Hsu[5], Masashi Miyauchi [2,6], Róbert Pálovics[3,7], Grace A. Meaker[5], Satoshi Kaito [8], Ola Rizq[8], Hwei Minn Khoo[5], Yavor Bozhilov[4,5], Kyomi J. Igarashi[2,6], Joydeep Bhadury [2,6,9], Christy Munson[3,7], Paul K. Mack[2,6], Tze-Kai Tan[1,2,6], Jan Rehwinkel[5], Atushi Iwama [8], Tony Wyss-Coray [3,7,10,11,15] ✉, Hiromitsu Nakauchi [2,6,12,15] ✉, Michael S. Haney[3,7,13,15] ✉ & Adam C. Wilkinson [4,5,15] ✉

The biological mechanisms that sustain the vast blood production required for healthy life remain incompletely understood. To search for cell intrinsic regulators of hematopoiesis, we perform a genome-wide in vivo hematopoietic stem and progenitor cell (HSPC)-based CRISPR knockout screen. We discover SAGA complex members, including *Tada2b* and *Taf5l*, as key regulators of hematopoiesis. Loss of *Tada2b* or *Taf5l* strongly inhibits hematopoiesis in vivo, causing a buildup of immature hematopoietic cells in the bone marrow. The SAGA complex deposits histone H3 lysine 9 acetylation (H3K9ac) and removes histone H2B ubiquitination (H2Bub). Loss of *Tada2b* leads to a reduction in H3K9ac levels and altered H2Bub enrichment in HSPCs, implicating disruption of SAGA complex activity. This is associated with upregulation of interferon pathway genes, reduced mitochondrial activity, and increased megakaryocyte progenitor cell commitment. Loss of these factors also enhances the cell outgrowth and the interferon pathway in an in vivo human myelodysplastic syndrome cell line model. In summary, this study identifies the SAGA complex as an important regulator of hematopoiesis.

Hematopoietic stem cells (HSCs) are a rare bone marrow (BM) cell type that have the potential to self-renew or undergo lineage commitment and differentiate into one of the many hematopoietic and immune cell types via a range of intermediate hematopoietic progenitor cells (HPCs)[1–5]. As such, hematopoietic stem and progenitor cells (HSPCs) support the hematopoietic and immune systems throughout life, which play various essential roles in human health from oxygen supply to wound healing to defense against pathogens. Approximately $10^{11}$ new blood cells are generated per day in humans. Hematopoietic system function generally declines with age and can result in leukopenia, anemia, myelodysplastic syndrome (MDS), and leukemia, as well

as other pathologies (e.g., atherosclerosis, osteoporosis)[6,7]. It is also one of the major drivers of declining immune cell function in aged individuals, predisposing aged individuals to infection and cancers[8]. Understanding the mechanisms of healthy and pathogenic hematopoiesis is therefore a key question in the field of hematology, stem cell biology, and gerontology. In particular, the regulation of gene expression is critical for healthy hematopoiesis[9,10]. However, there is still much to be explored about this process.

Tight regulation of gene expression is critical for HSPC function, with perturbations known to underlie the loss of hematopoietic system homeostasis[9]. The SAGA complex is an evolutionary-conserved

A full list of affiliations appears at the end of the paper. ✉e-mail: twc@stanford.edu; nakauchi@stanford.edu; michael.haney@pennmedicine.upenn.edu; acw63@cam.ac.uk

multiprotein complex that contains both histone acetyl-transferase and histone deubiquitinase activities[11,12]. The canonical enzymatic proteins are KAT2A and USP22, respectively. Loss of *Kat2a* has previously been shown to minimally impact HSC activity[13], perhaps due to redundancy with *Kat2b*, although it is important in leukemic stem cell activity[14]. Loss of *Usp22* has recently been shown to induce emergency myelopoiesis[15]. Alongside these enzymatic subunits, the SAGA complex contains several structural components including transcriptional adapters (TADAs) including TADA2B and TADA1, and transcription-associated factors (TAFs) including TAF5L and TAF6L. The role of these structural components in HSPC activity is poorly understood.

While HSPC-derived hematopoiesis is a well characterized system, the application of methods to identify genetic regulators of this process through high-throughput genetic screens have been hampered by the rarity of these cells. In vivo pooled CRISPR/Cas9 knockout (KO) screens offer an efficient method to investigate thousands of genetic regulators for a range of biological processes[16], and have already been applied to a number of cell lines[14,17,18]. However, the application of this system to primary HSPCs is considerably more challenging because of the large numbers of cells required to perform these screens[19]. To date, in vivo HSPC CRISPR screens have been limited to single guide RNA (sgRNA) libraries targeting 30–150 genes[20,21]. However, recent advances in polymer-based ex vivo HSC expansion now enable large numbers of HSPCs to be generated, albeit a heterogeneous mixture of HSCs and HPCs (with transplantable HSCs at a frequency of ~1:34)[22–24]. Here, we have utilized this ex vivo HSPC expansion methodology to undertake a genome-wide HSPC CRISPR screen, from which we identify and validate structural components of the SAGA complex as key regulators of hematopoiesis.

## Results

### In vivo CRISPR screen identifies SAGA complex members as hematopoiesis regulators

To gain insight into the genetics of hematopoiesis, we optimized a large-scale ex vivo HSPC expansion culture method[22–24] to generate sufficient HSPC numbers for CRISPR KO screens. We performed 10 pooled CRISPR KO screens targeting the entire mouse genome (10 sgRNAs/gene) in 3-week expanded primary mouse HSPC cultures. To identify molecular regulators of in vivo hematopoiesis, we transplanted these gene KO HSPCs into 65 lethally irradiated recipients (Fig. 1a), achieving high-level donor peripheral blood (PB) chimerism (Supplementary Fig. S1a). After 10–12-weeks, BM and spleen from these mice were collected pooled, and various hematopoietic cell types purified for sgRNA sequencing: c-Kit+Lineage- HSPCs, CD11b/Gr1+ myeloid cells, CD4/CD8+ T cells, B220+ B cells, and Ter119+ erythroid cells. We could detect an average of 5 sgRNA/gene for all genes in the mouse genome (Supplementary Fig. S1b, c).

In comparing sgRNA abundance between HSPCs and all mature lineage populations, gene hits included well-characterized hematopoiesis regulators such as *Runx1*, *Gata3*, and *Cic* (Fig. 1b, Supplementary Fig. S1f–l, Supplementary Data 1). Loss of the transcription factor *Runx1* in adult HSCs has been previously shown to inhibit lympho-poiesis while enhancing myelopoiesis[25]. In our pairwise analysis however, we observed relative loss of *Runx1* sgRNAs in all lineages including myeloid[26,27] (Supplementary Fig. S1f–i). To better understand this apparent discrepancy, we generated *Runx1* KO HSPCs via electroporation (Supplementary Fig. S1m, n) and transplanted them into lethally-irradiated recipients. Consistent with our screen results (Supplementary Fig. S1o), KO chimerism was higher within the c-Kit+Lineage- bone marrow than within the T-cell, B-cell and myeloid lineages (Supplementary Fig. S1p). However, also consistent with the literature, T-cell and B-cell chimerism was low while myeloid chimerism was high after 12-weeks (Supplementary Fig. S1q). These findings help to validate our in vivo HSPC CRISPR screen and provide confidence in the hits identified by it.

One striking result of the genome-wide screen was that we identified members of the SAGA complex[11] as being enriched within the HSPC population (Fig. 1b, c). These SAGA complex members (*Tada2b*, *Tada1*, *Taf6l*) were also seen as enriched when HSPCs were compared with each individual mature lineage (Supplementary Fig. S1f–i). To investigate this further, we performed a targeted ~2000 gene screen focused on regulators of gene expression that included many of the SAGA complex members. In this screen, we transplanted the gene expression KO library-containing HSPCs into 24 lethally irradiated recipients (Supplementary Fig. S1d, e). SAGA complex members were even more pronounced as top hits that are required for normal hematopoiesis in this focused screen (Fig. 1d–h, Supplementary Data 2).

### Loss of structural SAGA complex subunits block normal hematopoiesis

Focusing further on the SAGA complex, the screens identified a specific subset of unique structural components of the SAGA complex: *Tada2b, Tada1, Taf5l and Taf6l* (Supplementary Fig. S2a, b). The role of these components in hematopoiesis is poorly understood. Other SAGA complex members, including enzymatic subunits (*Kat2a*, *Kat2b*, *Usp22*) and those not unique to the SAGA complex (e.g., *Tada3* and *Sgf29*), were not identified as hits (Supplementary Fig. S2a, b). To validate these structural SAGA complex members as hematopoiesis regulators, we knocked out *Tada2b*, *Taf5l*, *Tada1*, or the *Rosa26* locus (as a control) in cultured HSPCs and transplanted them into recipient mice (Fig. 2a). Gene KOs were confirmed at the DNA and protein levels (Supplementary Fig. S2c–e). *Tada2b, Taf5l*, and *Tada1* KO cells all engrafted at much higher levels within the immature BM HSPC compartments as compared with the mature PB, while control sgRNA cells displayed similar chimerism levels throughout the hematopoietic hierarchy (Fig. 2b–g). We observed a striking increase in the frequency of immature hematopoietic cell populations (Lineage- cells and KSL HSPCs) within the KO BM cells (Fig. 2b, c). Notably, while higher levels of KO chimerism were observed in Lineage- progenitor populations, chimerism within the most primitive immunophenotypic CD150+CD34-KSL LT-HSC population was similar to the PB (Fig. 2d–g). Corresponding with the screen, SAGA component KO contribution to mature immune lineages was significantly reduced relative to HSPCs (Fig. 2h). Additionally, we confirmed that the inhibited hematopoiesis phenotype was transplantable by performing secondary transplantation assays (Supplementary Fig. S2f).

To confirm that these results are not confounded by extended time in ex vivo culture, we further validated that SAGA complex members altered the activity of functional HSCs in knock outs of *Tada2b* or *Taf5l* (using two different sgRNAs) in freshly-isolated CD150+CD34-c-Kit+Sca-1+Lineage- HSCs and evaluated their activity in 16-week transplantation assays. These experiments were performed with sgRNAs delivered through ribonucleoprotein (RNP) electroporation or with lentiviral transduction. Similar to our validation experiments, donor chimerism was again lower in the PB than in the immature KSL HSPCs from the *Tada2b*- and *Taf5l*-KO HSCs (Supplementary Fig. S2g–j). In the RNP KO transplantation assay, hematopoiesis was almost entirely donor-derived, we also observed significant reductions in white blood cell (WBC) counts in the *Tada2b*-KO and *Taf5l*-KO recipients (Supplementary Fig. S2k). Red blood cell (RBC) counts and platelet counts were not significantly reduced (Supplementary Fig. S2l–m). Together, these results confirmed SAGA complex members *Tada2b* and *Taf5l* as functional HSPC regulators that are required for normal hematopoiesis.

We next investigated the phenotype of *Tada2b*-KO and *Taf5l*-KO in our ex vivo HSPC cultures. Consistent with these knockouts displaying a block in differentiation, we observed an increase in the frequency of CD201+CD150+KSL population (described as the phenotypic (p)HSC population ex vivo that contains the functionally

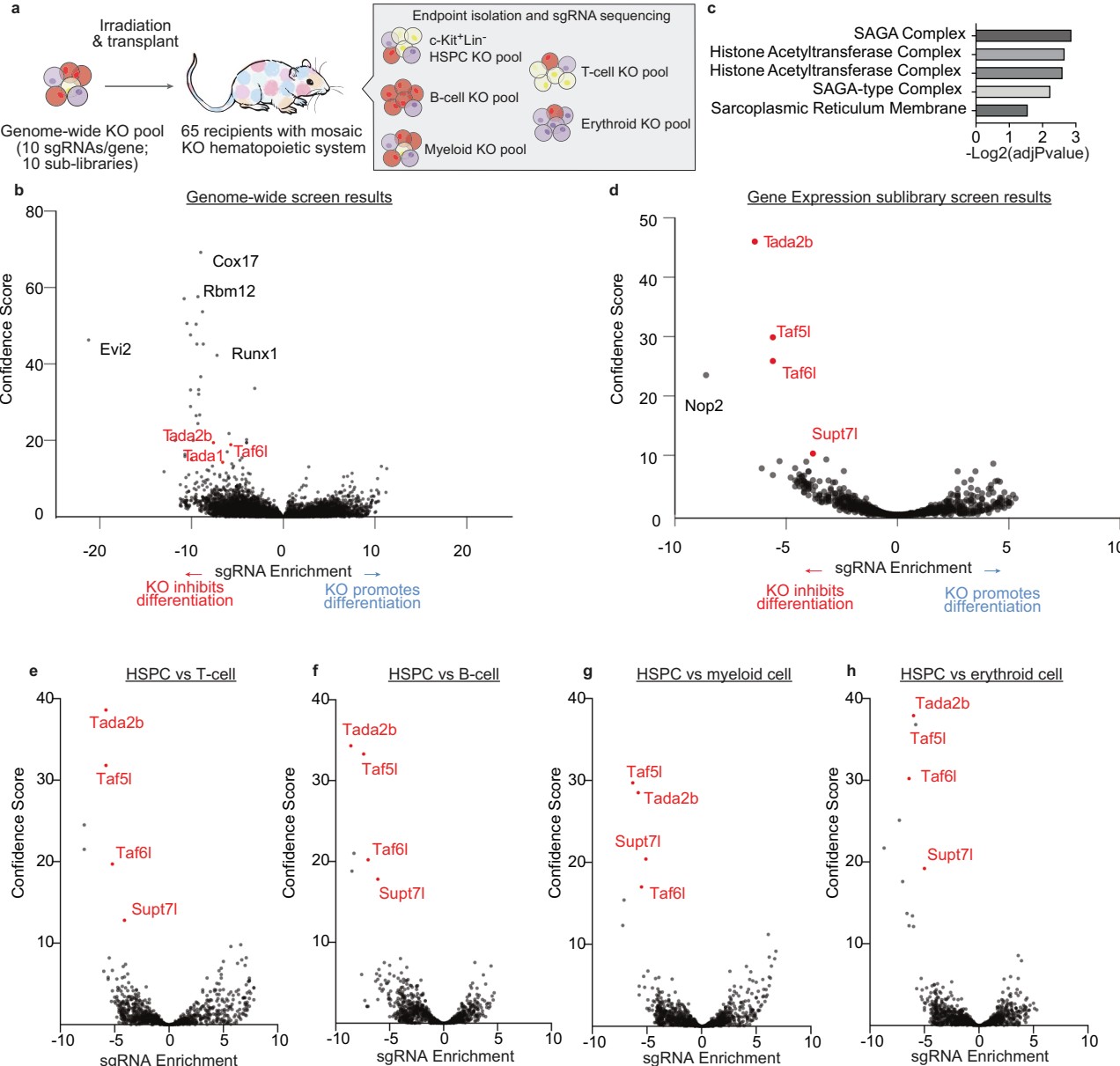

**Fig. 1 | In vivo HSPC CRISPR screen identifies SAGA complex members as putative regulators of hematopoiesis. a** Schematic of the in vivo genome-wide (GW) CRISPR knockout (KO) screen for hematopoiesis. **b** Volcano plot showing in vivo hematopoietic stem and progenitor cell (HSPC) CRISPR GW screen results comparing HSPC KOs with KO mature lineage cells. All KOs displayed with the effect of KO (negative if KO enriched in HSPCs, positive if KO enriched in mature lineage cells) on the x-axis and confidence score on the y-axis. SAGA complex members labeled in red. **c** Gene Ontology (GO) enrichment of top 200 screen hits ranked by confidence score. GO term enrichment p-values were computed using Fisher's exact test. Reported adjusted p-values correspond to Benjamini–Hochberg

FDR correction. **d** Volcano plot showing in vivo HSPC CRISPR screen results for sub-library of 2000 genes involved in gene expression. All KOs displayed with the effect of KO (negative if KO enriched in HSPCs, positive if KO enriched in differentiated cells) on the x-axis and confidence score is on the y-axis. SAGA complex members labeled in red. Volcano plot showing in vivo HSPC sub-library CRISPR screen results comparing c-Kit⁺Lineage⁻ HSPC KOs with KO mature T-cells (**e**), mature B-cells (**f**), myeloid cells (**g**), and erythroid cells (**h**). All ~2000 gene KOs displayed with the effect of KO (negative if KO enriched in HSPCs, positive if KO enriched in mature lineage cells) on the x-axis and confidence score on the y-axis. SAGA complex members labeled in red.

transplantable HSCs[28,29]) within the *Tada2b*-KO and *Taf5l*-KO cultures after 14-days (Fig. 3a). Numbers of pHSCs were not increased in these cultures however, suggesting this was due to a reduction in non-HSC populations rather than an increase in pHSC numbers (Supplementary Fig. S3a). We observed a loss of CD201⁻CD150⁻KSL and c-Kit⁺Sca-1⁻Lineage⁻ progenitor populations (Supplementary Fig. S3b, c). This pHSC expansion phenotype was also seen when quantifying c-Kit⁻Lin⁻CD201⁺ cell frequencies (Supplementary Fig. S3d). Additionally, knockdown of *TADA2B* using previously reported short-hairpin RNAs (shRNAs)[30] within ex vivo human HSPC cultures[31] induced a

similar increase in the frequency of immunophenotypic CD201⁺CD34⁺CD45RA⁻CD41⁻ HSC compartment (Fig. 3b). These results provided further evidence that loss of SAGA complex components altered HSPC activity and that we could investigate this phenotype ex vivo.

## HSPCs lacking SAGA complex members upregulate interferon pathways

Given these striking functional phenotypes from the genetic ablation of *Tada2b* and *Taf5l*, we were interested to better understand the

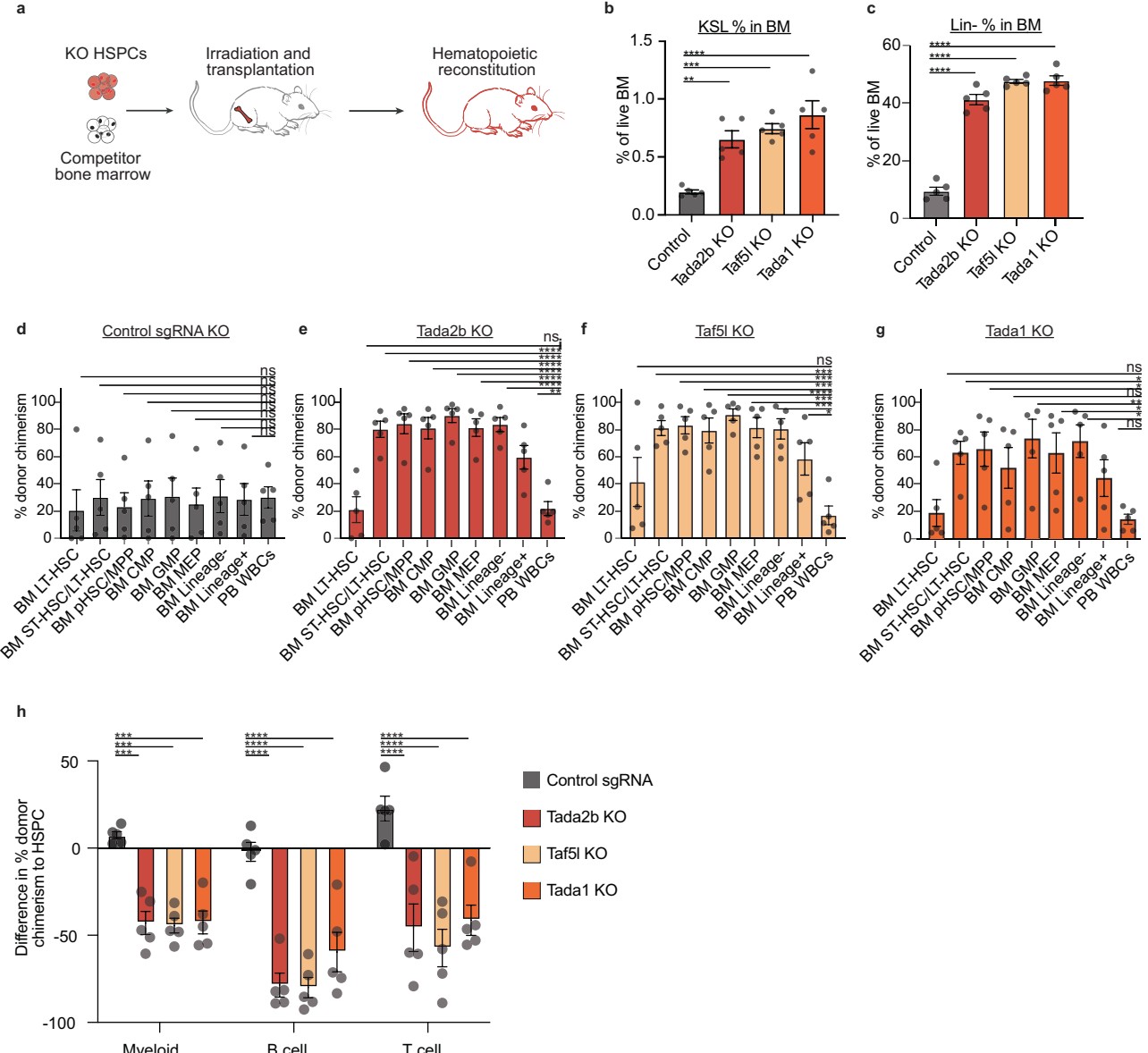

**Fig. 2 | Loss of SAGA complex members inhibits normal hematopoiesis.**
**a** Schematic for single gene KO validations. **b** Frequency of c-Kit⁺Sca-1⁺Lineage⁻ (KSL) cells within the CD45.1⁺ bone marrow (BM) compartment in transplant recipients at 12-weeks. *n* = 5 per condition. Error bars represent s.e.m; *P*-value determined by one-way ANOVA. ***P* = 0.0027; ****P* = 0.0005; *****P* < 0.0001. **c** Frequency Lineage⁻ cells within the CD45.1⁺ BM compartment in transplant recipients at 12-weeks. *n* = 5 per condition. Error bars represent s.e.m. *P*-value determined by one-way ANOVA. **P* < 0.05, *****P* < 0.0001. **d**–**g** Frequency of donor CD45.1⁺ chimerism

in indicated hematopoietic cell populations for *Rosa26*-KO cells (**c**), *Tada2b*-KO cells (**d**), *Taf5l*-KO cells (**e**), and *Tada1*-KO cells (**f**). *n* = 5 per condition. Error bars represent s.e.m; *P*-value determined by one-way ANOVA. **P* < .05, ***P* < .005, ****P* < .00005, *****P* < 0.0001. **h** Difference in donor chimerism between the KSL HSPC compartment and myeloid, T and B cells, for transplants described in (**a**). Error bars represent s.e.m; *P*-value determined by one-way ANOVA. ****P* < 0.0005, *****P* < 0.0001.

consequences of these gene knockouts at the molecular level. We therefore performed single cell RNA-sequencing (scRNA-seq) of control, *Tada2b*-KO, and *Taf5l*-KO HSPC cultures at 14-days post-RNP electroporation. Consistent with the known heterogeneity of our HSPC cultures, initial clustering analysis identified four major cell types including a population of cells with an HSC signature and several progenitor cell types (Fig. 3c, Supplementary Fig. S3e, f). This analysis suggested loss of *Tada2b* or *Taf5l* increased the frequency of megakaryocyte progenitor cells (MkPs) within the HSPC cultures (Supplementary Fig. S3g), which we could validate by flow cytometry (Supplementary Fig. S3h).

To investigate the molecular consequences specifically within the HSCs, we specifically subclustered on this population (Fig. 3c). By

combining this re-clustering with cell cycle analysis, we could identify sub-clusters resembling G1-phase HSCs, S-phase HSCs, and G2-phase HSCs (Fig. 3c). To avoid cell cycle-associated transcriptional changes, we focused on gene expression differences within the G1-HSC cluster. We observed major differences in the transcriptional programs expressed in the *Tada2b*- and *Taf5l*-KOs (Fig. 3d). Gene Ontology (GO) pathway analysis identified major changes in interferon (IFN) signaling (Fig. 3e, Supplementary Data 3-4). For example, interferon response genes *Ifi44*, *Ddx60*, and *Oasl2*, were significantly upregulated in the KO settings (Fig. 3f–h). Significant upregulation of IFN-related gene sets was also observed in other progenitor clusters (Supplementary Data 3-4). The similarity of these DEGs in the *Tada2b* and *Taf5l* KOs also implicate loss of SAGA complex activity as a common mechanism

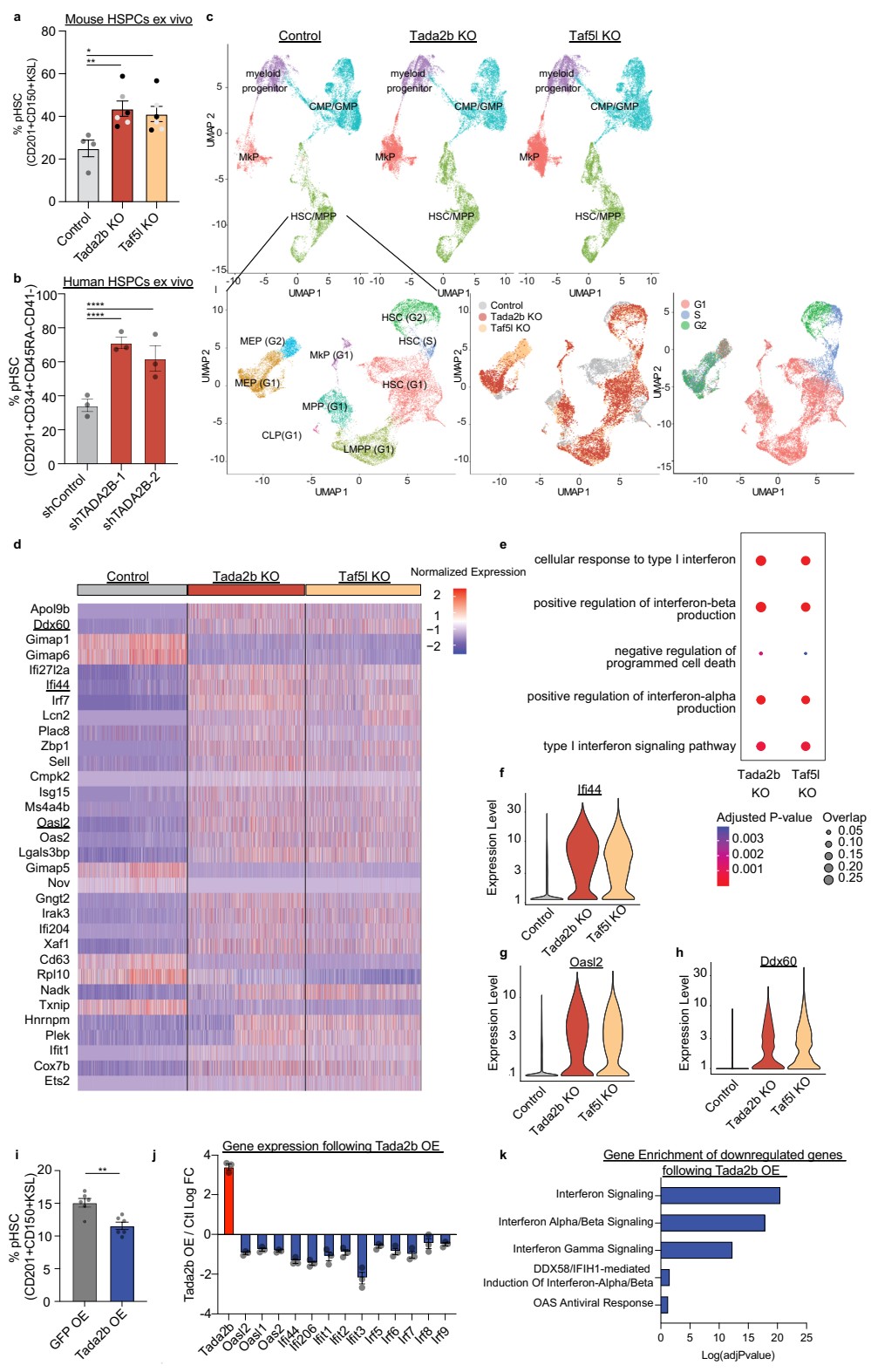

underlying these transcriptional changes. Corresponding with this gene signature, we observed protein level upregulation of known IFN response genes Sca-1 (Ly6e) and MHC-I (Supplementary Fig. S3i, j), although these did not reach statistical significance.

We were interested in whether similar gene expression changes were also induced in HSPCs in vivo. We therefore performed bulk RNA-seq on control, *Tada2b*-KO, and *Taf5l*-KO KSL HSPCs from the BM of

our transplantation recipients. We also observed increases in IFN response genes in this context, suggesting this phenotype was not an artifact of the HSPC culture system but was tightly linked to the loss of these SAGA components (Supplementary Fig. S4a–c, Supplementary Data 5). However, within the in vivo setting, we could not observe changes in Sca-1 (nor MHC-I) by flow cytometry (Supplementary Fig. S4d, e).

**Fig. 3 | Tada2b or Taf5l regulate the interferon pathway in HSPCs. a** Percentage of mouse phenotypic HSPCs (pHSPCs; CD201⁺CD150⁺KSL) in ex vivo cultures 14-days after ribonucleoprotein (RNP) KO of SAGA complex members. *n* = 4 for controls and *n* = 6 with two independent single guide RNAs (sgRNAs) used per gene KO (gray: sgRNA1, black: sgRNA2). Error bars represent s.e.m; *P*-value determined by one-way ANOVA. ***P* < 0.005; **P* < 0.05. **b** Percentage of human pHSPCs (CD201⁺CD34⁺CD45RA⁻CD41⁻) in ex vivo cultures 14-days after short-hairpin RNA (shRNA) knockdown of *TADA2B*. Three biological replicates displayed. Error bars represent s.e.m; P-value determined by one-way ANOVA. *****P* < 0.0005. **c** Above, UMAP representation of single cell RNA-seq of control, *Tada2b*-KO, and *Taf5l*-KO mouse HSPC cultures at day 14 with clusters annotated. Total of 61,324 single cells analyzed. Below, UMAP representations of the HSC/MPP cluster from a following sub-clustering with cluster, cell sample origin, and cell cycle state annotated (total of 21,344 single cells analyzed). CMP common myeloid progenitor, GMP granulocyte-macrophage progenitor, MkP megakaryocyte progenitor, HSC hematopoietic stem cell, MPP multipotent progenitor, MEP megakaryocyte-erythroid progenitor, CLP common lymphoid progenitor, LMPP lymphoid-primed multipotent progenitor. **d** Heatmap displaying the expression of the top 30 differentially

expressed genes ranked by adjusted *p*-value between the control, *Tada2b*-KO, and *Taf5l*-KO cultures within the G1-phase HSC cluster from (**c**). Rows represent sub-sampled single-cell gene expression values. Interferon (IFN) response genes are underlined. **e** GO enrichment analysis displaying adjusted *P*-value and gene-set overlap for 200 most differentially expressed genes between control and KO cells within the G1-phase HSPC cluster. Violin plots for the expression of IFN response genes *Ifi44* (**f**), *Oasl2* (**g**) and *Ddx60* (**h**) in control, *Tada2b*-KO and *Taf5l*-KO cells within the G1-phase HSC cluster. *P*-values determined by MAST. **i** Fold change of pHSC (CD201⁺CD150⁺KSL) frequency in ex vivo cultures with *Tada2b* or GFP overexpression (OE). Three independent replicates displayed. Error bars represent s.e.m; *P*-values determined by two-sided, paired *t*-test. ***P* = 0.0023. **j** Log fold change in expression of indicated genes in *Tada2b* OE HSPCs as determined by RNA-seq. *n* = 3 independent samples. Error bars represent s.e.m. **k** GO enrichment analysis displaying adjusted *P*-value for downregulated genes in *Tada2b* OE HSPCs. GO term enrichment *p*-values were computed using Fisher's exact test. Reported adjusted *p*-values correspond to Benjamini–Hochberg FDR correction.

To test whether SAGA complex members are not only necessary for restraining IFN signaling but sufficient for suppressing IFN genes, we overexpressed *Tada2b* in ex vivo expanded HSCs. We observed a decrease in CD201⁺CD150⁺KSL pHSC frequency in ex vivo HSPC cultures upon *Tada2b* overexpression (Fig. 3i), the opposite of the KO phenotype. *Tada2b* overexpression resulted in a suppression of several IFN-related genes in RNA-seq and was the top GO category in the set of significantly downregulated genes (Fig. 3j, k, Supplementary Fig. S4f, Supplementary Data 6), indicating that TADA2B expression plays an important role in regulating IFN gene expression in HSPCs.

To understand where the SAGA complex was influencing the intrinsic IFN pathway, we initially assessed inhibition of the upstream IFN pathway regulator TBK1. Inhibition led to ~50% reduction in the surface expression of the interferon-stimulated gene (ISG) Sca-1 (Fig. 4a). Similar effects were seen in *Tada2b*-KO and control backgrounds, suggesting that TBK1 activity was not unique to the *Tada2b*-KO setting. We next assessed whether IFN secretion (and signaling) could be contributing to the SAGA complex KO phenotype by performing an IFN secretion assay. We detected higher IFN levels in *Tada2b*-KO cultures as compared with control cultures (Fig. 4b).

To determine the contribution of secreted IFN(alpha/beta) on the SAGA complex KO-associated gene expression changes, we knocked out *Tada2b* in *Interferon alpha/beta receptor alpha chain* (*Ifnar1*)-deficient HSCs and performed bulk RNA-seq. The upregulation of IFN response genes was lost when *Tada2b* was knocked out in the *Ifnar1*-KO background (Fig. 4c–e), implicating IFNs as mediators of this gene expression pattern. However, another set of genes were similarly upregulated following loss of *Tada2b* in both the wildtype (WT) and *Ifnar1*-KO backgrounds (Fig. 4f, g). The increased CD201⁺CD150⁺KSL pHSC frequency phenotype was also retained in the *Ifnar1*-deficient background (Supplementary Fig. S4g, h). Additionally, the *Ifnar1*-KO background did not rescue the *Tada2b*-KO differentiation defect in vivo (Supplementary Fig. S4i). These results implicate IFN signaling as one of several molecular pathways dysregulated by the SAGA complex in HSPCs, in line with the multifaceted functions of the SAGA complex[11].

**Altered SAGA complex chromatin activity associated with loss of adapter proteins**

Loss of SAGA complex acetyltransferase enzymes (KAT2A and KAT2B) was recently shown to upregulate IFN signaling in the context of intestinal stem cells[32]. The study implicated mitochondrial dysregulation as a key driver of IFN secretion. We therefore quantified mitochondrial activity in *Tada2b*-KO HSPCs. We observed a modest but significant reduction in MitoTracker Orange CMTMRos (Fig. 5a), without a reduction in mitochondrial mass (quantified by MitoTracker

Green FM) (Supplementary Fig. S5a). Within the HSPC cultures, reduction in mitochondrial activity was slightly more pronounced in the pHSC fraction compared to total Lin⁻ cell population (Supplementary Fig. S5b). Consistent with reduced mitochondrial potential, *Tada2b*-KO HSPCs also displayed enhanced sensitivity to the mitochondrial Complex I inhibitor IACS-010759 (Fig. 5b).

To better understand the chromatin-level dysregulation caused by the loss of SAGA complex members in HSPCs, we next investigated the epigenetic marks mediated by the SAGA complex, histone H3 lysine 9 acetylation (H3K9ac) and histone 2B lysine 120 mono-ubiquitation (H2Bub). Loss of TADA2B, TAF5L, or TADA1 have recently been reported to destabilize the HAT domain in cell line models[33]. In line with this, we observed a ~20% loss of H3K9ac levels within *Tada2b*-KO HSPCs by intracellular flow cytometry (Fig. 5c), with similar results also seen in *Taf5l*- and *Tada1*-KO HSPCs (Supplementary Fig. S5c). By contrast, no significant change in H2Bub was observed in these KO HSPCs (Fig. 5d, Supplementary Fig. S5d). In line with a global loss of H3K9ac, we saw an overall reduction in chromatin accessibility in *Tada2b*-KO pHSCs by ATAC-seq (Fig. 5e, Supplementary Fig. S5e, Supplementary Data 7). Within the modest number of gained differentially accessible regions, we detected enrichment for IRF motifs (Fig. 5f) and interferon gene sets (Fig. 5g), which matches with the IFN expression pattern in our RNA-seq analysis.

To further probe the genome-specific changes in H3K9ac and H2Bub enrichment, we performed Chipmentation[34] in *Tada2b*-KO and WT pHSCs. Interestingly, we observed a significant shift in H2Bub enrichment (Fig. 5h, Supplementary Data 7), with a gain in H2Bub at genes associated with STAT and IRF4 motifs (Fig. 5i). These genes were enriched in pathways associated with interferon signaling and cholesterol metabolism (Fig. 5j), while H2Bub was lost at genes associated with cytokine signaling (Supplementary Fig. S5f). Despite the global loss of H3K9ac, we did not observe statistically significant changes in H3K9ac enrichment (Supplementary Fig. S5g), although we could observe reductions in the level of H3K9ac enrichment at several gene loci (e.g., *Kit* and *Stat5a*) (Supplementary Fig. S5j, k). Incomplete loss of H3K9ac peaks may be due to the incomplete KO of *Tada2b* in our HSPC cultures. Nonetheless, these results provide important insights into the chromatin-level consequence of loss of SAGA complex members in HSPCs.

To investigate whether *Tada2b*-KO induced loss of SAGA complex HAT activity was likely to be the major consequence, we tested the effect of GSK699, a KAT2A/B inhibitor/degrader[33]. Similar to *Tada2b*-KO HSPCs, GSK699 induced a reduction in H3K9ac levels relative to dimethyl sulfoxide (DMSO) control (Fig. 5k). No changes to H2Bub levels were observed (Fig. 5l). GSK699 also induced an upregulation of Sca-1 (Fig. 5m) and increased immunophenotypic HSCs and MkPs

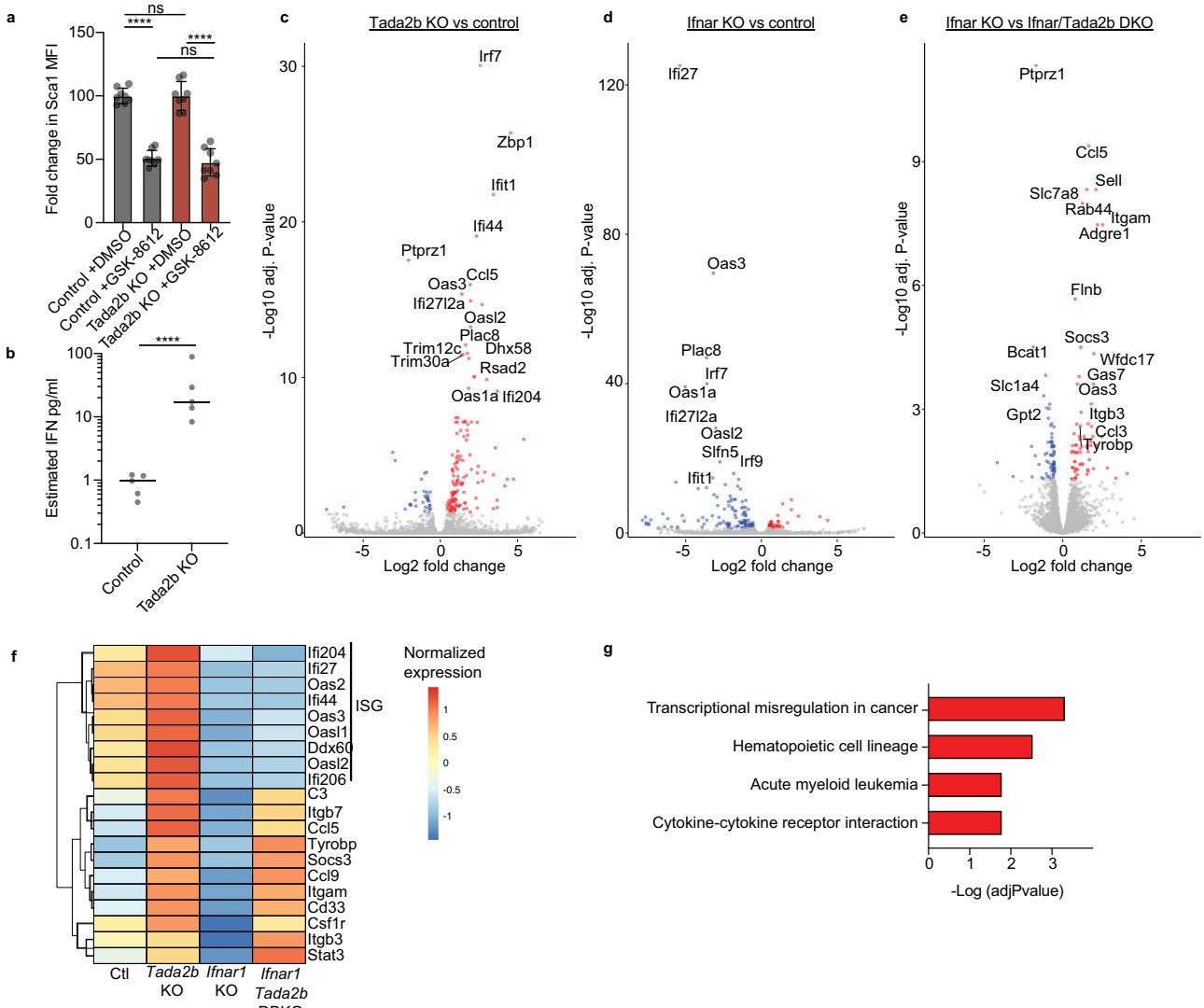

**Fig. 4 | SAGA complex members have IFN dependent and independent effects on HSPCs. a** Flow cytometric Sca-1 mean fluorescence intensity (MFI) in HSPC cultures following a 7-day treatment with the TBK1 inhibitor GSK-8612. $n = 8$ independent samples. Error bars represent s.e.m. *P*-values determined by one-way ANOVA. ****$P < 0.0005$. **b** Estimated concentration of IFNs in the supernatant of 14-day *Tada2b*-KO or control KO HSPC cultures. Supernatant collected 48 h after the last media change. $n = 5$ independent samples. *P*-values determined by two-sided, unpaired *t*-test. ****$P < 0.0005$. **c** Volcano plot comparing gene expression between control CD201$^+$CD150$^+$KSL HSCs and *Tada2b*-KO CD201$^+$CD150$^+$KSL HSCs, at 14-days culture after RNP electroporation. *P*-values determined by DESEQ2. Blue dots: significant negative log2 fold change; red dots: significant positive log2 fold change; gray dots: not significant. **d** Volcano plot comparing gene expression between control CD201$^+$CD150$^+$KSL HSCs and *Ifnar*-KO CD201$^+$CD150$^+$KSL HSCs, at

14-days culture after RNP electroporation. *P*-values determined by DESEQ2. Blue dots: significant negative log2 fold change; red dots: significant positive log2 fold change; gray dots: not significant. **e** Volcano plot comparing gene expression between *Ifnar1*-KO CD201$^+$CD150$^+$KSL HSCs and *Ifnar1*/*Tada2b* double-KO CD201$^+$CD150$^+$KSL HSCs, at 14-days culture after RNP electroporation. *P*-values determined by DESEQ2. Blue dots: significant negative log2 fold change; red dots: significant positive log2 fold change; gray dots: not significant. **f** Heatmap displaying normalized expression of indicated genes within indicated HSPC cultures. *P*-values determined by DESEQ2. DBKO double knockout, ISG Interferon-stimulated genes. **g** GO enrichment analysis for upregulated genes indicated in (**e**). GO term enrichment *p*-values were computed using Fisher's exact test. Reported adjusted *p*-values correspond to Benjamini–Hochberg FDR correction.

(Fig. 5n, o), in line with the *Tada2b*-KO. Bulk RNA-seq of c-Kit$^+$Sca-1$^+$Lin$^-$ HSPCs also confirmed a correlation in transcriptional changes induced by treatment with GSK699 and the loss of *Tada2b* (Fig. 5p, q, Supplementary Fig. S5l, Supplementary Data 8). The overlapping molecular phenotypes of GSK699 and *Tada2b*-KO suggest that TADA2B plays an important role in stabilizing the SAGA complex's HAT module in HSPCs.

**Loss of SAGA boosts engraftment and interferon signaling in a human MDS model**

The dramatic block in hematopoiesis was reminiscent of the failure seen in hematological diseases such as MDS. Additionally, MDS

samples are associated with IFN upregulation (Supplementary Fig. S6a) and mitochondrial dysfunction[35]. We therefore set out to evaluate whether loss of SAGA complex influenced the human MDS-L cell line model[36]. We knocked out SAGA complex members *TADA2B*, *TAF5L*, *TADA1*, or the *AAVS1* control locus in MDS-L cells using CRISPR (Supplementary Fig. S6b). We then performed a competitive transplantation assay into immunodeficient mice expressing human IL-3 and GM-CSF (Fig. 6a). We observed a striking competitive advantage by the SAGA component KO MDS-L cells as compared with the control cells, with significantly higher chimerism in both the BM and spleen (Fig. 6b, c, Supplementary Fig. S6c). Additionally, we observed larger spleen weights in *TADA1*- and *TAF5L*-KO recipients (Supplementary Fig. S6d).

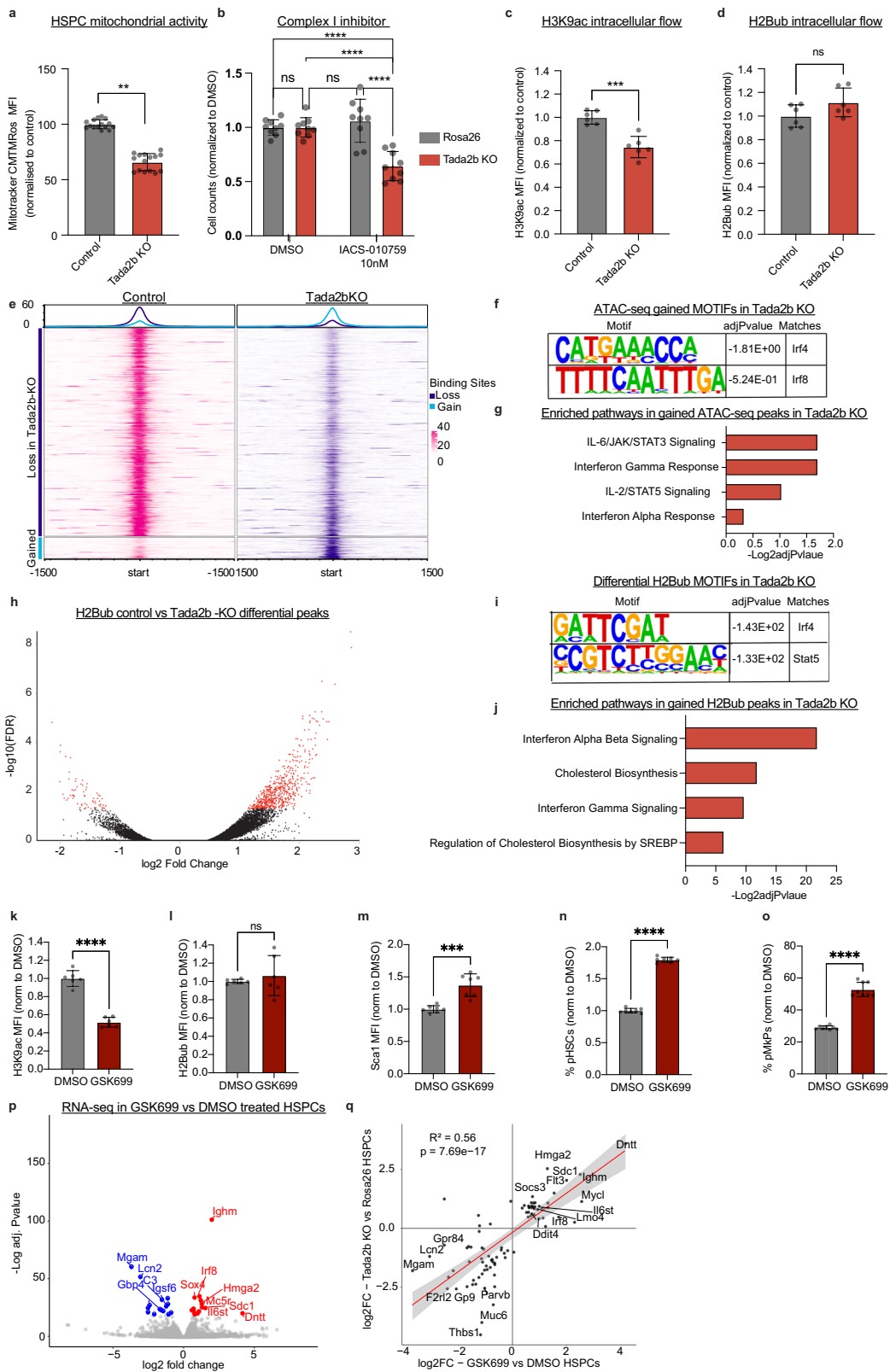

However, there were no overt signs of leukemic transformation (Supplementary Fig. S6e) or recipient mortality.

To see if the same transcriptomic signature were present in the human SAGA complex KOs as in the mouse setting, we performed RNA-seq on the 12-week BM engrafted human SAGA complex KOs MDS-L cells as well as controls. We again observed a robust upregulation of IFN response genes upon SAGA complex member KOs (Fig. 6d, e, Supplementary Data 9). In addition to these IFN genes, we also observed an upregulation of myeloid gene sets including *CSF3R* and *CD33* (Fig. 6e, f). Together, these results suggest the SAGA complex has various regulatory activities that are conserved between mouse and human.

**Fig. 5 | Altered SAGA complex chromatin activity associated with loss of adapter proteins. a** MitoTracker Orange CMTMRos signal in control (*Rosa26*-KO) in *Tada2b*-KO HSPCs. *n* = 14 independent replicates. Error bars represent s.e.m; *P*-values determined by two-sided, paired *t*-test. \*\**P* < 0.005. **b** Cell counts following a 5-day treatment with Complex I inhibitor (IACS-010759) or dimethyl sulfoxide (DMSO) on control or *Tada2b*-KO HSPCs. *n* = 9 independent replicates. *P*-value determined by one-way ANOVA. Error bars represent s.e.m. \*\*\*\**P* < .0005. (**c**) Intracellular flow cytometry for H3K9ac in control and *Tada2b*-KO HSPCs. *n* = 6 independent replicates. Error bars represent s.e.m; *P*-values determined by two-sided, paired *t*-test. \*\*\**P* < .0005. **d** Intracellular flow cytometry for H2Bub in control and *Tada2b*-KO HSPCs. *n* = 6 independent replicates. Error bars represent s.e.m; *P*-values determined by two-sided, paired *t*-test. **e** Profile plot of ATAC-seq data representing ATAC-seq reads gained or lost in *Tada2b*-KO relative to control HSPCs. Data averaged from two independent samples. **f** HOMER based motif enrichment analysis of differentially gained ATAC-seq peaks in *Tada2b*-KO HSPCs compared to controls. Motif enrichment was assessed using HOMER. Enrichment p-values were calculated using a cumulative binomial test comparing motif frequencies in target versus background sequences. **g** Reactome analysis of genes annotated near differentially gained ATAC-seq peaks in *Tada2b*-KO HSPCs compared to control HSPCs. GO term enrichment *p*-values were computed using Fisher's exact test. Reported adjusted *p*-values correspond to Benjamini−Hochberg FDR correction. **h** Volcano plot of differential peak analysis of H2Bub Chipmentation sequencing of *Tada2b*-KO HSPCs compared to controls. Points in red represent FDR < .05. *P*-values were determined by DiffBind using DESeq2-based differential analysis of read counts. **i** HOMER based motif enrichment analysis of differentially gained H2Bub Chipmentation sequencing peaks in *Tada2b*-KO HSPCs compared to controls. Motif enrichment was assessed using HOMER. Enrichment p-values were calculated using a cumulative binomial test comparing motif frequencies in target versus background sequences. **j** Reactome analysis of genes annotated near differentially gained H2Bub Chipmentation sequencing peaks in *Tada2b*-KO HSPCs compared to control HSPCs. GO term enrichment p-values were computed using Fisher's exact test. Reported adjusted *p*-values correspond to Benjamini−Hochberg FDR correction. Intracellular flow cytometry for H3K9ac (**k**) or H2Bub (**l**) in wildtype (WT) HSPCs following a 48-h treatment with a KAT2A/B inhibitor (GSK699) or DMSO. *n* = 6 independent replicates. Error bars represent s.e.m; *P*-values determined by two-sided, unpaired *t*-test. \*\*\*\**P* < 0.0005. **m** Flow cytometric Sca-1 MFI expression in WT HSPCs following a 48-h treatment with a KAT2A/B inhibitor (GSK699) or DMSO. *n* = 7 independent replicates. Error bars represent s.e.m; *P*-values determined by two-sided, unpaired *t*-test. \*\*\**P* < 0.005. Frequency of immunophenotypic HSCs (pHSCs, in **n**) or immunophenotypic MkPs (pMkPs, in **o**) in WT HSPC cultures following a 48-hour treatment with a KAT2A/B inhibitor (GSK699) or DMSO. *n* = 7 (**n**) and *n* = 8 (**o**) independent replicates. Error bars represent s.e.m; *P*-values determined by two-sided, unpaired *t*-test. \*\*\*\**P* < 0.0005. **p** Volcano plot representing differential gene expression in WT HSPCs following a 48-h treatment with a KAT2A/B inhibitor (GSK699) or DMSO. *P*-values determined by DESEQ2. **q** Scatter plot representing top 50 DEGs induced by a KAT2A/B inhibitor (GSK699) and KO of *Tada2b*. *P*-values determined by DESEQ2.

## Discussion

In this study, we identified and validated structural components of SAGA complex members, TADA2B, TAF5L, and TADA1, as regulators of hematopoiesis. Loss of these factors led to buildup of immature HSPCs in the bone marrow, and a loss of mature blood cells in the peripheral blood following transplantation. At the transcriptional level, this was associated with an upregulation in IFN response genes in HSCs ex vivo and in vivo, likely due to the increased secretion of IFN. However, loss of IFNa/b sensitivity did not entirely reverse the SAGA component KO HSPC phenotype, suggesting additional mechanisms of the SAGA complex in HSPCs that warrant further investigation. In line with a recent study in intestinal stem cells, loss of SAGA complex activity resulted in reduced mitochondrial activity. Reduced mitochondrial activity is associated with defect hematopoiesis. However, further work is needed to confirm that the hematopoiesis defects induced by SAGA complex KOs were mediated by mitochondrial dysregulation or additional molecular mechanisms.

At the chromatin level, loss of these SAGA complex members led to a global reduction in H3K9ac. In line with a key role for loss of H3K9ac, several molecular and cellular features of the *Tada2b*-KO HSPC cultures could be recapitulated by treatment with a KAT2A/ KAT2B inhibitor/degrader (GSK699). A recent study observed TADA2B, TAF5L, and TADA1 as key regulators of KAT2A stability (and activity), with loss resulting in SAGA complex instability and KAT2A degradation. Notably, neither *Kat2a* nor *Kat2b* were identified as hits in our screen, suggesting that one can replace the other's activity in HSPCs. Similar enzymatic redundancy of KAT2A and KAT2B has been observed in other systems, in line with this hypothesis. However, we also identified large changes in H2Bub enrichment, suggesting the SAGA complex DUB activity may also be altered. Further work will be needed to determine whether these are direct or indirect effects of SAGA disruption.

Alongside its contribution to HSPC activity, we also identified a role for these SAGA complex members in a human MDS model, where loss of these SAGA complex members enhanced outgrowth in vivo. Upregulation of IFN response genes was also seen in this human cell context, suggesting a conserved molecular pathway in mouse and human. However, somatic mutations in SAGA complex members were infrequent in primary leukemia samples (Supplementary Fig. S6f). Nonetheless, SAGA complex activity in this context requires future investigation. Given the functional studies of the SAGA complex members we describe here, we hypothesize that loss of SAGA complex activity might be contributing to age-related dysfunction of hematopoiesis. Additionally, our findings suggest that these structural components warrant study in other adult stem cell systems.

This study also highlights the potential for combining genetic screens with recently developed HSPC expansion approaches. The applications of this HSPC CRISPR screening approach are abundant; from identifying regulators of self-renewal and hematopoiesis, to investigating mechanisms in the full range of biological functions of mature HSPC-derived immune cells or searching for synthetic lethality gene interactions in (pre-)malignant stem cells.

### Limitations of study

While we hope that our in vivo HSPC CRISPR screen will offer a valuable resource for the research community, we acknowledge the limitations of the current study. First, to generate the sufficient number of HSPCs for these large-scale genetic screens, we needed to expand the HSPCs ex vivo prior transplantation. We acknowledge that certain cellular characteristics might be changed by ex vivo culture (and transduction) and this culture period might be driving skews in the sgRNA distribution targeting genes that regulate HSC survival and growth. Second, we transplanted a heterogeneous mixture of ex vivo expanded HSPCs, which engraft and reconstitute hematopoiesis at different rates. Third, we acknowledge incomplete recovery of the sgRNA libraries in vivo. This is likely driven at least in part by this biological heterogeneity but also by loss of sgRNA detection due to the inefficiency of cell recovery from the bone marrow. Recent improvements in barcoding sgRNA libraries should help to improve future in vivo screens[37].

## Methods
### Mice
Mice were housed in a 12 h dark-light cycle, ambient temperature (20−22 °C), and humidity-controlled (30−70%) environment with ventilated caging, sterile bedding, and unrestricted access to sterile food and water in the animal facilities at Stanford University or at the University of Oxford. C57BL/6-Rosa26$^{CAG-Cas9}$ (B6J.129(Cg)-Gt(ROSA) 26Sor$^{tmL1(CAG-cas9*,-EGFP)Fezh}$/J; 026179) and C57BL/6-CD45.1 (PepboyJ; 002014) were purchased from The Jackson Laboratory. C57BL/6-CD45.2 mice purchased from The Jackson Laboratory (000664) or bred at the University of Oxford. C57BL/6-CD45.1/CD45.2 F1 mice were bred from C57BL/6-CD45.1 and C57BL/6-CD45.2 at Stanford University

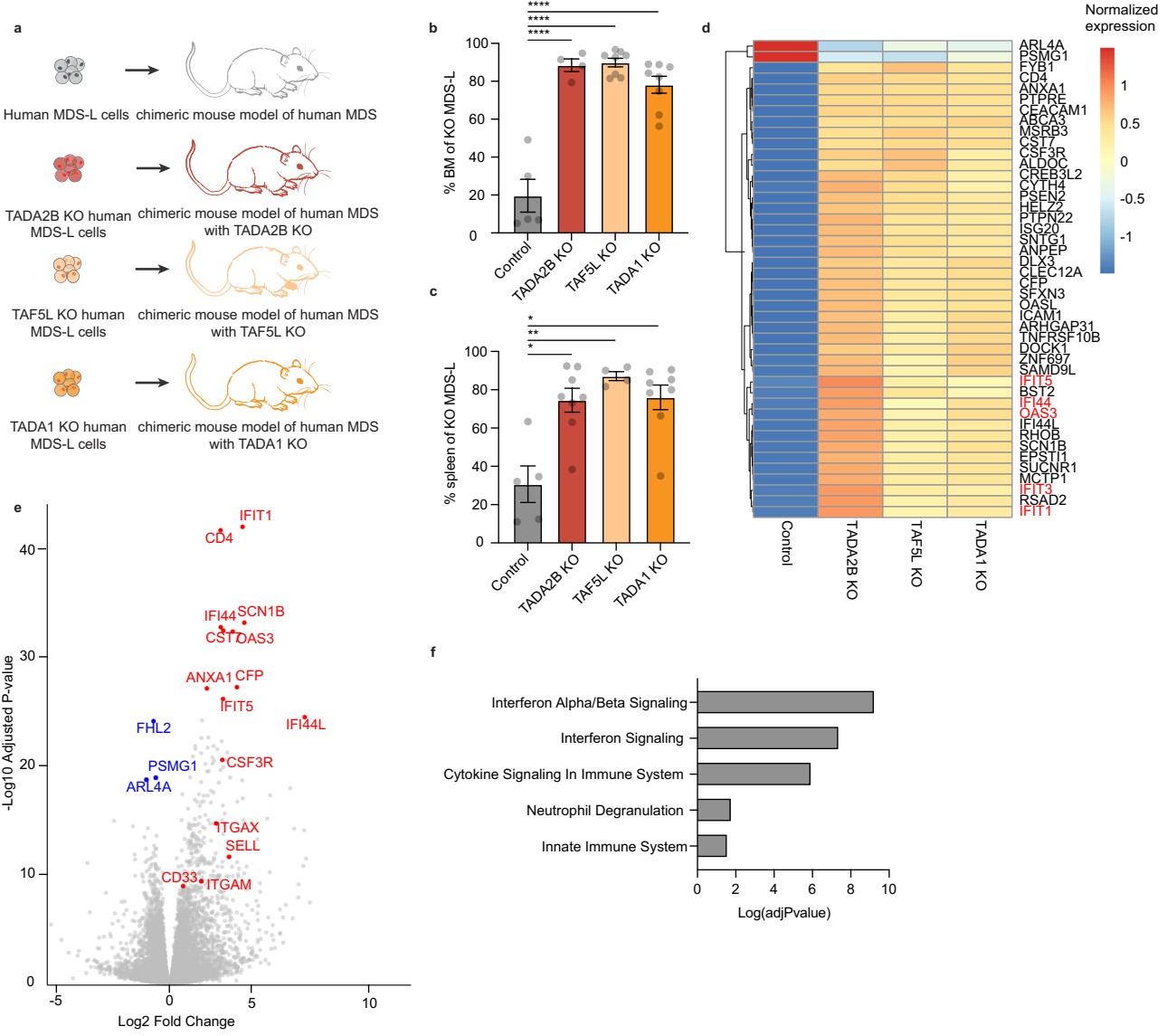

**Fig. 6 | SAGA complex members regulate cell outgrowth in myelodysplastic syndrome. a** Human MDS-L cell chimeric mouse model schematic. **b** Frequency of KO MDS-L cells in BM. *n* = 5 independent replicates. *P*-value determined by one-way ANOVA. Error bars represent s.e.m. \*\*\*\**P* < 0.0005. **c** Frequency of KO MDS-L cells in spleen. *n* = 5 independent replicates. *P*-value determined by one-way ANOVA. Error bars represent s.e.m. \*\**P* < 0.005, \**P* < 0.05. **d** Heat map comparing gene expression

between in vivo MDS-L control cells with in vivo MDS *TADA2B*-KO, *TAF5L*-KO and *TADA1*-KO cells. **e** Volcano plot comparing gene expression between in vivo MDS-L control cells with in vivo MDS *TADA2B*-KO cells. *P*-values determined by DESEQ2. **f** GO enrichment of upregulated genes in MDS-L *TADA2B*-KO cells as compared with control cells. GO term enrichment *p*-values were computed using Fisher's exact test. Reported adjusted *p*-values correspond to Benjamini–Hochberg FDR correction.

or the University of Oxford. NOD.Cg-Prkdc$^{scid}$ Il2rg$^{tm1Sug}$ Tg(SRa-IL3, CSF2)7-2/Jic (NOG-EXL) mice expressing human IL-3 and GM-CSF (NOG/IL-3/GM-CSF) were purchased from In-Vivo Science Inc. *Infar1*-KO (B6.bkg-Ifnar1$^{tm1Agt/Mmjax}$) mice were provided by Caetano Reis e Sousa. All mice were 8–12 weeks at the experiment start point.

### Ex vivo mouse HSC cultures

Mouse HSC cultures were initiated from c-Kit$^+$ bone marrow[38] from C57BL/6-Rosa26$^{CAG-Cas9}$ mice. Following $CO_2$ euthanasia, tibia, femur, pelvis, and vertebrae were collected and crushed to release bone marrow cells. Bone marrow was then stained with anti-cKit-APC antibody (clone 2B8, Biolegend), then incubated with anti-APC microbeads (Miltenyi) and c-Kit$^+$ cells enrichment was performed using LS MACS column (Miltenyi) according to the manufacturer's protocol. Enriched c-Kit$^+$ bone marrow cells were cultured in HSC expansion media at 5% $CO_2$ and 5% $O_2$ in a CellBind coated cell culture plate (Corning, cat. CLS3337) with complete media changes every 2-3 days.

HSC expansion media composition[24,39]: Ham's F12 media supplemented with 1X penicillin-streptamycin-glutamine (PSG; Gibco), 1X HEPES, 1X insulin-transferrin-selenium-ethanolamine, 1 mg/ml polyvinyl alcohol (Sigma), 100 ng/ml recombinant mouse thrombopoietin (Peprotech), and 10 ng/ml recombinant mouse stem cell factor (Peprotech). Detailed information on all products used are listed in the Supplementary Table 1. For validation experiments, HSC cultures were initiated with c-Kit$^+$ bone marrow from C57BL/6-CD45.1, C57BL/6-CD45.2, or *Ifnar1*-KO mice, as indicated in the text. Where indicated, HSC cultures were supplemented with the TBK1 inhibitor GSK-8612 at 5 µM or DMSO. Where indicated, HSPCs were cultured for 100 nM GSK699, 10 nM IACS-010759 or equivalent volume of DMSO.

### Ex vivo and in vivo CRISPR screen

After 3-week ex vivo expansion of c-Kit$^+$ bone marrow HSPCs, ~50 million cells were transduced overnight with lentivirus generated from one of 10 Bassik mouse CRISPR knockout sub-libraries[40] (a kind gift

from the Bassik lab; Addgene 1000000121-1000000130). Each sub-library contained ~20,000 elements. Lentivirus was generated using helper plasmids pCMV-VSV-G-RSV-Rev and pMDLg/pRRE. To reduce multiple integrations per cell, we used a transduction efficiency of 20–30%, to achieve 500–750× coverage (10–15 million transduced HSPCs per sub-library). Two days later, transduced HSPCs were selected for using puromycin for 72 h. HSPCs were recovered and expanded in culture for another 9 days. For ex vivo timepoints, 20 million HSPCs were harvested for DNA extraction. At day-14 post-transduction, HSPCs were transplanted into lethally-irradiated (10 Gy) C57BL/6-CD45.1 recipients (1–2 million HSPCs per recipient; >10 million per sub-library across 6–8 mice). A total of 65 mice were transplanted for the genome-wide screen, while 24 mice were transplanted for the Gene Expression-focused screen. After 10–12-weeks, whole bone marrow and spleen were harvested. From the whole bone marrow, c-Kit$^+$Sca-1$^+$Lineage$^-$ or c-Kit$^+$Lineage$^-$ cells were isolated by FACS. The remaining bone marrow and spleen cells were pooled and CD45R$^+$ B cells, CD4/CD8$^+$ T cells, Mac1/Gr1$^+$ myeloid cells, and Ter119$^+$ erythroid cells were isolated via MACS columns.

## Screen analysis

Genomic DNA was extracted for all screen populations separately according to the protocol included with QIAGEN Blood Mini or Midi Kit. Using known universal sequences present in the lentivirally incorporated DNA (oMCB 1562 and oMCB 1563), sgRNA sequences were amplified and prepared for sequencing by two sequential polymerase chain reactions as previously described[40]. Products were sequenced using an Illumina Nextseq to monitor library composition (20–40 million reads per sample). Trimmed sequences were aligned to libraries using Bowtie, with zero mismatches tolerated and all alignments from multi-mapped reads included. Guide composition and comparisons across each cellular fraction were analyzed using CasTLE version 1.0[40]. Briefly, enrichment of individual sgRNAs was calculated as a median-normalized log ratio of the fraction of counts. For each gene, a maximum likelihood estimator was used to identify the most likely effect size and associated log-likelihood ratio (confidence score) by comparing the distribution of gene-targeting guides to a background of nontargeting and safe-targeting guides. P-values were then estimated by permuting gene-targeting guides, and hits were called using FDR thresholds calculated via the Benjamini-Hochberg procedure.

## Ex vivo validation assays

To generate *Taf5l* or *Tada2b*-KO HSCs, $2 \times 10^5$ HSCs derived from two-week C57BL/6 c-Kit$^+$ bone marrow (as described in the Ex vivo HSC culture section) were electroporated with 6 µg recombinant HiFi Cas9 (IDT) pre-complexed with 3.2µg synthetic sgRNA (Synthego; sequences listed in the Supplementary Table 1) using a Lonza 4D-Nucleofector using program EO100 in complete primary P3 solution (16.4 µl of P3 primary cells solution + 3.6 µl supplement per reaction) and then returned to HSC expansion media[41]. Electroporation of Cas9 only or *Rosa26*-targeting RNP were used as controls. At indicated timepoints, cells were collected for KO frequencies or flow cytometry analysis. To determine KO frequencies, gDNA was isolated via QuikExtract for PCR (primer sequences listed in the Supplementary Table 1) and Sanger sequencing with KO frequencies determined using ICE software (Synthego). Alternatively, lentivirus carrying Tada2b-2A-GFP cDNA, or GFP cDNA were generated and transduced into 3-week old Cas9-expressing (or wild-type) C57BL/6 HSC cultures, as described in the Ex vivo and in vivo CRISPR screen section. Flow cytometric analysis was then performed following antibody staining (CD201-APC, CD150-PE/Cy7 cKit-BV421, Sca-1-PE, Gr1-FITC, Ter119-FITC, CD4-FITC, CD8-FITC, CD45R-FITC, CD127-FITC, all diluted 1:500 in PBS) for 30 min at 4 °C, then washed and analyzed using an LSRFortessa (BD) with propidium iodide (Biolegend) as a live/dead cell stain. All flow cytometric data

analysis was performed using FlowJo and statistical analysis performed using Prism.

## Flow cytometric analysis

For HSPC culture flow cytometric analysis, cells were antibody stained (CD201-APC, CD150-PE/Cy7, cKit-BV421, Sca-1-PE, Gr1-APC/eFluor780, Ter119-APC/eFluor780, CD4-APC/eFluor780, CD8-APC/eFluor780, CD45R-APC/eFluor780, CD127-APC/eFluor780, all diluted 1:500 in PBS, staining volume 100ul per sample) for 30 min at 4 °C, washed and then analyzed using a LSRFortessa (BD) with propidium iodide (Biolegend) as a live/dead cell stain. Peripheral blood samples were analyzed using a Horiba Micros ESV 60 complete blood counter or by flow cytometry. For flow cytometry, peripheral blood leukocytes were stained with antibodies (CD45.1-PE/Cy7 1:400, CD45.2-BV421 1:400, Mac1-PE 1:2000, Gr1-PE 1:2000, CD45R-APC/Cy7 1:1000, CD4-APC 1:2000, CD8-APC 1:2000 in PBS, staining in 100ul volume) following red blood cell lysis. Samples were then run on an FACSAriaII or LSRFortessa using propidium iodide as a live/dead cell stain, with CD45.1$^+$ cells sorted for KO analysis. Data analysis was performed using FlowJo. Bone marrow chimerism was also analyzed by sequential antibody staining with a lineage cocktail (Gr1-biotin, Ter119-biotin, CD4-biotin, CD8-biotin, CD45R-biotin, CD127-biotin, all 1:33 in PBS, staining volume 100ul per sample) followed by Streptavidin-APC/eFluor780 1:100, CD34-FITC 1:25, c-Kit-APC 1:100, Sca-1-PE 1:100, CD45.1-PE/Cy7 1:33, CD45.2-BV421 1:33, CD16/32-BV711 1:100, MHC I-AF700 1:300 OR Streptavidin-APC/eFluor780 1:100, CD34-FITC 1:25, c-Kit-APC 1:100, Sca-1-PE 1:100, CD45.1-PE/Cy7 1:33, CD45.2-BUV395 1:33, CD48-BV421 1:100, CD150-BV785 1:100 in PBS, staining volume 100ul per sample). Samples were then run on an FACSAriaII or LSRFortessa using propidium iodide as a live/dead cell stain, with CD45.1$^+$c-Kit$^+$Lineage$^-$ cells sorted for KO analysis. For intracellular staining, cells were fixed and permeabilized using the BD Transcription Factor Buffer set and stained with an anti-Ubiquityl-Histone H2B (Lys120)-PE or anti-Acetyl-Histone H3 (Lys9)-PE antibodies for 45 min at 4 °C (both diluted 1:100 in the Perm/Wash buffer from the kit, staining in 100 µl per sample), then washed and analyzed. For the assessment of mitochondrial mass and mitochondrial activity, cells were stained with MitoTracker Green FM (1:1000 in PBS from stock prepared according to the manufacturer's instructions, staining in 100 µl volume) or MitoTracker Orange CMTMRos (final concentration 30 nM in PBS) dyes, respectively for 45 min at 37 °C according to the manufacturer's recommendations, then washed and analyzed. All flow cytometric data analysis was performed using FlowJo and statistical analysis performed using Prism. Antibody information is listed in the Supplementary Table 1. Immunophenotypic definitions of the cell populations described in this manuscript are listed in Supplementary Table 2 below and Supplementary Fig. S7a–d.

## Western blotting

Protein lysates extracted from control and KO HSCs were run on a 4–12% Bis-Tris gel and blotted using antibodies against TADA2B (St. John's Laboratory, STJ194361-200), TAF5L (Proteintech, 19274-AP), TADA1 (Proteintech, 20337-1-AP), GAPDH (Bethyl Laboratories, A300-641A), or histone H3 (Abcam, ab1791). Following secondary staining with peroxidase-conjugated goat anti-rabbit IgG (Sigma Aldrich, A6667), blots were imaged using a ChemiBlot Imager and the Precision Plus Protein Kaleidoscope kit (Biorad 1610375).

## Transplantation assays using RNP-KO HSCs

CD150$^+$CD34$^-$Kit$^+$Sca-1$^+$Lineage$^-$ HSCs were isolated using a FACSAriaII (BD) from C57BL/6-CD45.1 bone marrow and electroporated with either Cas9/sgRNA targeting *Tada2b*, *Taf5l*, or *Rosa26* (same protocol as Ex vivo validation assays section) and then expanded for 7-days before transplantation into lethally-irradiated (10 Gy) C57BL/6-CD45.2 recipient mice (500 HSC equivalent per recipient) alongside 1 million whole bone marrow competitor cells isolated from C57BL/6-CD45.1/

CD45.2 F1 mice. At the endpoint, bone marrow and spleen were collected and fixed (4% paraformaldehyde for 48 h) for hematoxylin and eosin staining (performed by the Stanford Animal Histology Service).

## Transplantation assays using RNP-KO HSPCs
C57BL/6-CD45.1 c-Kit⁺ bone marrow from were expanded for 14-days and then electroporated with either Cas9/sgRNA targeting *Tada2b*, *Taf5l*, *Tada1*, *Runx1* or *Rosa26* (same protocol as **Ex vivo *validation assays*** section). After a second 14-day culture, cells were transplanted into lethally-irradiated C57BL/6-CD45.2 recipient mice (5000 CD201⁺CD150⁺c-Kit⁺Sca-1⁺Lin⁻ cells per recipient) alongside 1 million whole bone marrow competitor cells isolated from C57BL/6-CD45.1/CD45.2 F1 mice. Peripheral blood (PB) analysis was performed as described in the *Flow cytometric analysis* section. BM and spleens were harvested at the endpoint of the experiment. For secondary transplantation assays, 5 million whole bone marrow cells from primary recipients were transplanted into lethally-irradiated recipient mice and analyzed as above.

## Interferon secretion assay
NIH/3T3 cells were incubated with HSC culture supernatant (48 h after the last media change) or with recombinant IFNa (dilution series from 100 ng/ml to 10 pg/ml; Biolegend, 752802) for 3.5 h and then collected for RNA extraction using the Quick-RNA microprep kit (Zymo Research, R1055). Following cDNA synthesis using the SSIV Reverse Transcriptase (ThermoFisher, 18090050), Taqman qPCR was run to quantify Ifit1 (ThermoFisher, Mm00515153_m1) and Gapdh (ThermoFisher, Mm99999915_g1) expression.

## Human shRNA knockdown assay
Four-day cultured human umbilical cord blood CD34⁺ HSPCs (originally purchased from NHSBT) were transduced with lentivirus carrying control or *TADA2B* targeting shRNAs (reported previously[30]) and were then cultured for 14 days. HSPCs were cultured in 3A media conditions[31]: IMDM supplemented with Butzyamide (Cellaid), 740 Y-P (MedChemExpress), UM729 (StemCellTechnologies), and FLT3L (Peprotech). HSPCs were then stained with CD34-APC/Cy7, CD41-APC, CD45RA-BV785, and CD201-PE and analyzed by flow cytometry using PI as a live/dead cell stain.

## Human MDS-L studies
Human mScarlet-expressing MDS-L cells[36] (a kind gift from Tohyama laboratory, Kawasaki Medical School) were generated previously[42] and maintained in RMP1640 supplemented with 10% fetal bovine serum, 1X PSG, and 10 ng/ml human IL-3 (Peprotech). MDS-L/mScarlet/Cas9 cells were initially generated using pLentiCas9-Blast lentivirus (Addgene 52962). MDS-L/mScarlet/Cas9 cells were then transduced and Puromycin-selected with indicated pMCB306 lentivirus expressing sgRNA and GFP (a kind gift from the Bassik lab; Addgene 89360). After confirming high KO efficiency, a 1:1 mixture of MDS-L/Cas9 cells and MDS-L/mScarlet/Cas9/sgRNA cells were then transplanted into NOG mice expressing human IL-3 and GM-CSF. After 12-weeks, bone marrow and spleen were harvested from recipient mice for flow cytometric analysis.

## Bulk RNA-seq
RNA was extracted from indicated cell populations using the Zymo Direct-zol RNA MicroPrep kit and RNA sequencing was performed by Nocogene using polyA enrichment and sequencing on a Novoseq X (20 M reads/sample). Reads were mapped to the mouse GRCm38 (mm10) reference genome using STAR (v.2.5.1b). Raw read counts were generated with STAR using the GeneCounts function. Differential expression in RNA-seq was analyzed using the R package DESeq2 (Love et al. 2014).

## ATAC-seq
HSCs were pelleted at $400 \times g$ and 4 °C. Next, pellets were gently resuspended in ice-cold 50 µl of lysis buffer (10 mM Tris-HCl pH 7.4, 10 mM NaCl, 3 mM MgCl₂, 0.1% IGEPAL CA-630) and spun down at $500 \times g$ for 10 min and 4 °C. The supernatants were discarded and pellets gently resuspended in 50 µl of transposition reaction mix (25 µl of tagment DNA buffer (Nextera, Illumina), 2.5 µl of tagment DNA enzyme (Nextera, Illumina), 22.5 µl of nuclease free water) and incubated at 37 °C for 30 min. Tagmented DNA was purified using MinElute PCR purification kit (Qiagen) and size selected for 70–500 base pairs (bp) using AmpureXP beads (Beckman Coulter). Libraries were constructed and amplified using 1.25 µM Nextera index primers and NEBNext High-Fidelity 2× PCR Master Mix (New England BioLabs). A quantitative PCR was run to determine the optimal number of cycles. Libraries were gel size selected for 165–300 bp fragments. Sequencing was carried out using an Illumina NexSeq 550, paired-end, 2× 100 bp. Reads were mapped to the mm10 genome using Bowtie2, and SAM files were converted to BAM format with Samtools. The BAM files were sorted and indexed using Samtools, and mapping statistics were generated. PCR duplicates were removed with Samtools, and BED files were created using bedtools bamtobed. Reads mapping to chrM and regions in the ENCODE mm10 blacklist were removed from the BED files. Genome coverage was calculated using bedtools genomecov to produce bedGraph files. A scale factor was calculated by dividing 5,000,000 by the number of regions in each BED file, and applied to the bedGraph files. BigWig files were generated using bedGraphToBigWig. Peaks were called with MACS2 in broad mode. Differential peak analysis was performed using DiffBind, and differential peaks were annotated and analyzed for motif enrichment using HOMER.

## Chipmentation
CD201⁺CD150⁺KSL cells were fixed with 1% formaldehyde (10 minutes incubation at room temperature with rotation), then washed in PBS. Cell pellets were then resuspended in 120 µl of lysis buffer (50 mM Tris pH8, 10 mM EDTA, 0.5% SDS, 1.2 µl PIC per every 120 µl in ddH₂O), and sonicated in Covaris AFA fiber microtubes (Covaris, cat. 520045) aiming for 200 bp fragments. SDS was neutralized with 1% Triton (25 µl of 10% Triton X100 for every 225 µl of sample), and samples were incubated at room temperature for 10 min with rotation. 5 µl of fresh Protein A dynabeads (ThermoFisher, cat. 10001D, washed with PBS to remove the residual storage buffer), were added per reaction, followed by incubation at 4 °C for 30 min with rotation. Samples were then placed on a magnet and cleared chromatin was collected, 5% of each reaction was allocated to a new DNA LoBind tube and stored at −20 °C as input control. Protein A dynabeads (ThermoFisher 10001D, 10 µl per antibody per sample) were washed twice with the wash buffer (200 µL PBS with 0.5% BSA + 1 µl PIC per antibody per sample), and resuspended in 200 µl wash buffer per reaction. Beads were then transferred to fresh DNA LoBind tubes and 1 µl of antibody was added to each reaction. The beads were then incubated for 3–4 h at 4 °C with rotation. The beads were then washed twice in a new wash buffer (500 µl PBS + 0.5% FCS + 1 µl Tween, 250 µl for each wash), 250 µl of cleared chromatin was added to each antibody-bead tube followed by an overnight incubation at 4 °C with rotation. The following day each reaction was washed three times with 150 µl of RIPA buffer (50 mM HEPES pH 7.6, 50 mM LiCl, 1 mM EDTA, 1% NP-40, 0.7% Na Deoxycholate in ddH₂O), once with 150 µl of TE buffer, and once with 150 µl of 10 mM Tris pH 8. The beads were then re-suspended in 25 µl of tagmentation buffer (prepare 5X stock of 50 mM Tris pH8, 25 mM MgCl₂ and 50% dimethylformamide in ddH₂0, and dilute to 1X with ddH₂0) pre-warmed to 37 °C, followed by addition of 5 µl of diluted Tn5 transposase (Illumina, cat. 20034197, 1 µl of Tn5 per 4 µl of tagmentation buffer) and incubation at 37 °C for 5 min with rigorous mixing. The beads were then immediately washed once with the RIPA

buffer, once with 10 mM Tris pH8 (150 µl per tube) and resuspended in ddH$_2$O (22.5 µl per tube). For input controls, 2.5 µl 50 mM MgCl2 was added to 12.5 µl of input chromatin sample, followed by the addition of 6 µl of 5× tagmentation buffer, 1 µl of Tn5 and 8 µl of ddH$_2$O; samples were then incubated at 37 °C for 10 min (mixed after 5 min), placed on ice and 20 µl were taken for library preparation step. For library preparation, the beads were supplemented with 25 µl of 2X NEBNext Ultra II Q5 Mastermix (New England Biolabs, cat. M0544S), 1.25 µl of universal adapter primer and 1.25 µl of index adapter primer and subjected to the following PCR programme: 72 °C 5 min, 95 °C 5 min, 12 cycles of (98 °C 10 s, 63 °C 30 s, 72 °C 3 min), 72 °C 5 min, 12 °C indefinite hold. Upon library preparation PCR, the tubes were transferred to the magnet and supernatant was put in fresh DNA LoBind tubes. 50 µl of the AMPure XP beads (Beckman Coulter, cat. A63881) were added to each tube, and the samples were incubated at room temperature for 2 min. The beads were then washed twice with 150 µl of freshly prepared 80% ethanol, incubated at room temperature for 2 min, ethanol was removed and the beads were air-dried, resuspended in 11 µl ddH$_2$O and incubated at room temperature for 2 min again. The tubes were put on the magnet and 10 µl were collected from each tube. QC and quantification of the samples were performed using D1000 high sensitivity ScreenTape (Agilent, cat. 5067-5584) and KAPA Library Quantification Kit (Roche, cat. KK4824). Library amplification was performed using KAPA HiFi HotStart Library Amp Kit (Roche, cat. KK2612) according to manufacturer's recommendations.

Raw reads were assessed for quality using FASTQC and adapters were trimmed with Trimmomatic. Reads were mapped to the mm10 genome using Bowtie2, and SAM files were converted to BAM format with Samtools. The BAM files were sorted and indexed using Samtools, and mapping statistics were generated. PCR duplicates were removed with Samtools, and BED files were created using bedtools bamtobed. Reads mapping to chrM and regions in the ENCODE mm10 blacklist were removed from the BED files. Genome coverage was calculated using bedtools genomecov to produce bedGraph files. A scale factor was calculated by dividing 5,000,000 by the number of regions in each BED file, and applied to the bedGraph files. BigWig files were generated using bedGraphToBigWig. Peaks were called with MACS2 in both broad and narrow modes. Differential peak analysis was performed using DiffBind, and differential peaks were annotated and analyzed for motif enrichment using HOMER.

### Single-cell RNA-seq analysis
At day-14 post-electroporation, control, *Tada2b*-KO, or *Taf5l*-KO HSC cultures were collected for RNA-seq analysis using a Gene Expression kit v3 (10× Genomics) according to the manufacturer's protocol with libraries sequenced on a NovaSeq 6000 S4 (Illumina). Raw gene counts were obtained by aligning reads to the mouse genome using CellRanger software (v.4.0.0) (10× Genomics). For quality control, cells with unique feature counts over 2500 or less than 200 were removed as well as cells that have >5% mitochondrial counts. Ambient cell free mRNA contamination was removed using SoupX for each individual sample. Multiplets were filtered out using DoubletFinder. The core statistical parameters of DoubletFinder (nExp and pK) used to build artificial doublets for true doublet classification were determined automatically using recommended settings. The SCTransform-based integration workflow of Seurat was used to align data, using default settings. In brief, the integration workflow searches for common gene modules (anchors) in cells with similar transcriptomes. Individual samples after undergoing quality control are integrated in a stepwise fashion, using cellular sequencing depth as a covariate to mitigate technical artifacts. After combining the samples into a single dataset or Seurat object, genes were projected into principal component space using the principal component analysis (RunPCA). The first 100 dimensions were used as inputs into the FindNeighbours, FindClusters

and RunUMAP functions of Seurat. In brief, a shared-nearest-neighbor graph was constructed based on the Euclidean distance metric in principal component space, and cells were clustered using the Louvain method. RunUMAP functions with default settings were used to calculate 2D UMAP coordinates and search for distinct cell populations. Distinct cell populations were annotated based on conical marker genes. Monocle3 (v.0.2.1.) was used to generate the pseudotime trajectory analysis of HSCs. Cells were re-clustered as described in and used as input into Monocle to infer cluster and lineage relationships within a given cell type. Specifically, UMAP embeddings and cell subclusters generated from Seurat were converted to a cell_data_set object using SeuratWrappers (v.0.2.0) and then used as input to perform trajectory graph learning and pseudotime measurement through reversed graph embedding with Monocle. Cell cycle analysis was performed in Seurat using a list of cell cycle markers from Tirosh et al.[43]. Differential gene expression was performed in Seurat using the MAST package under default conditions.

### Ethics
All animal experiments were approved by the Administrative Panel on Laboratory Animal Care at Stanford University, the UK Home Office, or the Animal Care and Use Committee of the Institute of Medical Science University of Tokyo. Human primary cell experiments were approved by the University of Oxford's research ethics committee (OxTREC-574-23).

### Reporting summary
Further information on research design is available in the Nature Portfolio Reporting Summary linked to this article.

## Data availability
Data supporting the findings of this work are available within the paper and its Supplementary Raw datasets and materials generated and analyzed during the current study are available via NCBI's Gene Expression Omnibus (GEO) under accession numbers GSE312143 (CRISPR screen), GSE312125 (scRNA-seq), GSE312282 (bulk RNA-seq), GSE312290 (ATAC-seq), GSE312126 (ChIP-seq). The genome-wide CRISPR screen data are available via an interactive web application: www.hematopoiesiscrisprscreens.com. Source data are provided with this paper.

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

## Acknowledgements

We thank the Stanford Stem Cell Institute FACS Core and WIMM Flow Cytometry Core for flow cytometry access, the Stanford Animal Histology Service for histology, the CZ-Biohub for performing the next generation sequencing, and the WIMM Genome Engineering Core for plasmid cloning. We thank Cristina Pina for providing shRNA plasmids and advice. We thank Amanda Ghassaei for assistance with the interactive webapp and figures. A.C.W. acknowledges support from the Kay Kendall Leukaemia Fund (KKL1378), the European Hematology Association (RG-202211-02958), the Wellcome Trust (302479/Z/23/Z, 203151/Z/16/Z, 203151/A/16/Z), the NIH (K99HL150218), and the Leukemia and Lymphoma Society (3385-19). H.N. was supported by the NIH (R01DK116944; R01HL147124), the Ludwig Foundation, the Luducq Foundation, and the Japan Society of the Promotion of Science. J.B. was supported by an international postdoc grant (2017-0034) from the Swedish Research Council and the Assar Gabrielsson Foundation, Sweden. M.S.H. acknowledges support from the Buck Institute and NIA (T32AG000266). T.W.C. acknowledges support from the NIH (AG072255). This project was supported by an IMSUT Collaborative Research Grant (K22-1073).

## Author contributions

A.S., L.O., I.H., M.M., R.P., G.A.M., S.K., O.R., H.M.K., Y.B., K.J.I., J.B., C.M., P.K.M., T.K.T., J.R., A.I., M.H., and A.C.W. performed experiments and analyzed data. A.S., L.O., H.N., T.W.C., M.H., and A.C.W. wrote the manuscript. All authors reviewed and edited the manuscript.

## Competing interests

H.N. is a co-founder of Celaid Therapeutics. A.C.W. is a consultant for ImmuneBRIDGE. However, none of these companies had input into the design, execution, interpretation, or publication of the work in this manuscript. No other authors declare competing interests.

## Additional information

[1]Graduate Program in Stem Cell Biology and Regenerative Medicine, Stanford University School of Medicine, Stanford, CA, USA. [2]Institute for Stem Cell Biology and Regenerative Medicine, Stanford University School of Medicine, Stanford University, Stanford, CA, USA. [3]Wu Tsai Neurosciences Institute, Stanford University, Stanford, CA, USA. [4]Department of Haematology, Cambridge Stem Cell Institute, University of Cambridge, Cambridge, UK. [5]MRC Weatherall Institute of Molecular Medicine, University of Oxford, Oxford, UK. [6]Department of Genetics, Stanford University School of Medicine, Stanford, CA, USA. [7]Department of Neurology and Neurological Sciences, Stanford University School of Medicine, Stanford, CA, USA. [8]The Institute of Medical Science University of Tokyo, Tokyo, Japan. [9]Department of Laboratory Medicine, Institute of Biomedicine, Sahlgrenska Academy, University of Gothenburg, Gothenburg, Sweden. [10]ChEM-H, Stanford University, Stanford, CA, USA. [11]Paul F. Glenn Center for the Biology of Aging, Stanford University School of Medicine, Stanford, CA, USA. [12]Stem Cell Therapy Division, Institute of Integrated Research, Institute of Science Tokyo, Tokyo, Japan. [13]Department of Pathology and Laboratory Medicine, University of Pennsylvania, Philadelphia, PA, USA. [14]These authors contributed equally: Archana Shankar, Leonid Olender. [15]These authors jointly supervised this work: Tony Wyss-Coray, Hiromitsu Nakauchi, Michael S. Haney, Adam C. Wilkinson. ✉e-mail: twc@stanford.edu; nakauchi@stanford.edu; michael.haney@pennmedicine.upenn.edu; acw63@cam.ac.uk

