## [Transparent Peer Review file · Nature Communications]

In vivo CRISPR screening identifies SAGA complex members as key regulators of hematopoiesis

Corresponding Author: Dr Adam Wilkinson

Version 0:

Reviewer comments:

Reviewer #1

(Remarks to the Author)

In this manuscript, Haney and colleagues present a genome-wide in vivo CRISPR-Cas9 knockout screen in mouse "hematopoietic stem cells", which identifies structural components of the SAGA complex, particularly Tada2b and Taf5l, as regulators of hematopoiesis. The authors showed ex vivo and in vivo that knockout of these genes results in a block in differentiation from early stem/progenitor cells. Molecular profiling reveals upregulation of interferon-stimulated genes following knockout in the hematopoietic stem/progenitor cell compartment, and loss of these genes enhances outgrowth in a myelodysplastic syndrome model.

The approach presented in this study could be enormously enabling for the field of hematopoietic research, given that the scale of screens that can be conducted using the ex vivo methods described here, could enable many further screens. Moreover, the phenotypes observed are of potential interest. However, there are some limitations to the work as presented. Specifically, the hits from the screen appear insufficiently validated, there is ambiguous interpretation of the phenotypes, and a lack of significant mechanistic insight based on these findings. I detail these specific concerns below.

Major Concerns:

1. The robustness of the screen is unclear. HSPCs were transduced with the library with limited coverage and subjected to extensive ex vivo expansion prior to transplantation, during which functional changes by knockout of critical factors could introduce significant bias. Although controlled subsequently, this likely impacts the power of this screen. Positive controls (e.g. known stem cell regulators) should be evaluated and discussed to confirm the validity of the screen. The authors also compared HSPCs versus individual lineages, are the well-established TFs for lineage commitment enriched in the screen?
2. The term "HSCs" is inconsistently applied to both ex vivo cultured HSPCs and phenotypic HSCs. More broadly, the manuscript is ambiguous about the specific stages of differentiation impacted by knockout of the targets in validation experiments. This needs to be more clearly discussed and delineated throughout.
3. The SAGA complex contains multiple modules. Although prior studies on other subunits are briefly summarized in the introduction, it is perplexing to see that the other components, particularly the catalytic subunits, were not included in the focused CRISPR screen or experimentally explored. Without addressing these, claims about SAGA complex function remain overstated.
4. Related to the previous point, given the well-established role of the SAGA complex in transcriptional regulation, the impact of knockouts on chromatin biology and gene regulation relevant to hematopoiesis should be explored. At a minimum, given the extensive literature in this field, how gene expression changes observed in this setting relate to other perturbations involved in gene regulation that impact HSCs should be examined. Some of this can be achieved by examining publicly available datasets.
5. (sc)RNA-seq results showed upregulation of IFN genes, but the causal relationship between this signature and the observed phenotype as well as the unchanged key markers in vivo remain largely underdeveloped. This requires further development, as discussed in point 4 above.

6. The use of an MDS cell line appears a little arbitrary as there is no evidence presented that the candidate genes are mutated, dysregulated, or otherwise implicated in MDS. Can better rationale for this exploration be presented?

Minor Concerns:

1. Figures need revision-1c, 3d, and 4f lack proper labeling, 1e uses incorrect nomenclature for mouse genes, 2g used inconsistent color schemes.

Reviewer #2

(Remarks to the Author)

Reviewer #3

(Remarks to the Author)

In this manuscript Haney et al describe an in vivo CRISPR/Cas9 screen to evaluate genes required for normal hematopoietic differentiation. They develop an ex vivo culture system to maintain HSCs allowing expansion and lentiviral transduction, followed by transplantation into lethally irradiated recipient mice and an in vivo readout. Using this elegant system they identify components of the SAGA complex as mediators of hematopoietic differentiation. They show that loss of these components leads to upregulation of the interferon response mouse hematopoietic cells. Finally, they show that knockout of these genes in the MDS-L model promotes engraftment and growth of these cells in a xenograft model. The in vivo screening system is exciting, and the role of SAGA in hematopoiesis is intriguing. Overall, the paper is clearly written, although lacking somewhat in discussion, but there is a lack of clarity as to what SAGA's role in hematopoietic differentiation is. I lay out these concerns plus some additional points below:

-The details of the whole-genome screen are not clearly described. Was each sub-pool of the library used to infect 50 million HSCs and then transplanted into 65 recipients? Or is the 50 million cells and 65 mice the total for the entire screen (all the sub pools combined)? Was the screen done in singlicate or replicate? Having specific numbers and a clearer description is helpful in assessing the representation and potential coverage of the screen. The schematic in Figure 1a also does not contain much detail or add much in explaining how the screen was performed. One of the more novel and exciting aspects of this report is the whole-genome in vivo screening approach, so additional details are important.

-1-2 million HSCs were transplanted per mouse, however generally it is thought that many fewer HSC's actually contribute to adult hematopoiesis (maybe around 100,000), so it's unclear how this would impact representation within the screen. Did the authors evaluate sgRNA representation in the pooled HSC's prior to transplantation, or in either the HSC or mature cell pools? If possible, some measure of this should be included in the supplemental methods/data.

-"Here we refer to the cells in these cultures as HSCs; we acknowledge that these cultures include hematopoietic progenitor cells but for simplicity we will refer to these as HSC cultures." Why not refer to these as hematopoietic stem and progenitor cells (HSPCs), which would be more accurate, and the abbreviation is just as easy to use throughout the paper? I feel that this term is more accurate and will avoid potential confusion.

-Were these the only 4 members of the SAGA complex to score in the screens? What about other members of the complex or the catalytic subunit, KAT2A?

-It's a little unclear what is being proposed as a model for the function of SAGA in hematopoiesis. There are potentially multiple different SAGA complexes or modules with different component members. Is the hypothesis that the components of SAGA complex identified in this screen make up a particular complex important for directly regulating a subset of genes important for hematopoiesis? Are there published ChIPseq data for KAT2A or other SAGA complex members that might shed light on which of these effects is direct?

-The focus here is on SAGA-mediated regulation of the interferon response. However, it is not clear that this is mediating the hematopoietic phenotype. Is there a hematopoietic phenotype to the *Ifnar1* knockout? Did any component of the interferon response score in the original genome-wide screen? What is mediating the hematopoietic differentiation defect?

-What is the SAGA target that mediates the hematopoietic differentiation defect? Can the authors cross-reference the list of genes or pathways up- or down-regulated in the SAGA mutant cells with the genes that scored in the genome-wide screen? This might provide insight into the mechanistic role that SAGA plays in hematopoiesis.

-The authors make the case that knockout of SAGA complex members supports the development of MDS/AML. To my knowledge, mutations in SAGA complex members have not been reported in clinical cohorts of MDS/AML. Are there any data to suggest that there is up- or down-regulation of these genes during disease formation, or anything else to suggest that changes in SAGA function participate in disease biology in humans?

Minor issues:

-Page 2, Line 65-67 (and throughout the manuscript): Mouse genes should have the first letter capitalized, and the remainder should be lower case letters and then italicized. Mouse protein names should have all their letters capitalized. So, "Kat2a" should be KAT2A. See https://www.informatics.jax.org/mgihome/nomen/short_gene.shtml for further reference.

-Figure 1e-h: The gene names are capitalized here, but should be lower case.

-Figure 3d: the legend says, "IFN response genes are highlighted in red." However, no genes are highlighted in red in the figure.

-Figure 4a: The figure suggests that there is a highly significant statistical difference between the "Control +TBKi" and "Tada2b KO +TBKi" conditions. However, by eye these two conditions look identical. Can the authors confirm whether this is correct or not?

-Methods describing analysis of bulk RNAseq are not included in the methods section.

Reviewer #4

(Remarks to the Author)

Haney et al performed a genome-wide CRISPR KO screen in ex vivo expanded murine cKit⁺ cells and following transplantation identified gene edits corresponding to members of the SAGA complex that were enriched in HSPCs vs mature cells in vivo. The results were confirmed using a focused CRISPR screen targeting around 2000 genes. The authors focus on Tada2b and Taf5l KO and show an accumulation of ST-HSC/progenitors in bone marrow (but not LT-HSCs). Leveraging single cell transcriptomics in CD201⁺CD150⁺KSL (ex vivo cultures), the authors show activation of IFN genes in KO cells compared to controls. Over expression of Tada2b dampened these signatures. Increased IFN secretion from cultured KO Sca1⁺ cells, reduced numbers of Sca1⁺ cells following GSK-8612 and Tada2b edits in ifnar KO cells showed elevation of alternate sets of genes (that maintained elevation of HSCs) but not of IFN genes. Gene edits were then created in MDS-L cells and accumulation of these in xenograft BM/Spleen and activation of IFN gene sets in harvested cells. This is an elegant manuscript with well-designed experiments and clear writing. The mouse experiments in particular are very well done.

Figure 1: Was this a single expansion culture and transplant of cells divided across 65 recipients with material pooled and sorted to identify sg enriched in stem/progenitors? If so, the schematic could be made clearer. How many recipient mice were used for the 2000 gene (SAGA enriched) screen?

Figure 2: Please show the flow gating strategy used for immunophenotyping. Particularly for the BM LT-HSCs vs ST/LT-HSCs and pHSC/MPPs (Figure 2 c-f) given the difference in donor chimerism (LT-HSCs being similar PB WBCs).

Figure 3:

- Phenotypic KO LT-HSCs expanded ex vivo (Fig 3a/128-130) but KO LT-HSCs were lower in vivo? Showing total cell expansion numbers rather than frequencies and flow cytometry gating will be helpful (ditto 3i and S4g)
- Comparisons were limited to HSCs in G1- was the IFN signature also increased in MPPs in G1?
- Mkp numbers appear to be increased in Tada2b and Taf5l KO?

Overall, there is a lot of cell type nomenclature throughout this paper (MPP/HSC, pHSC, LTHSC, STHSC, Mkp etc) with very little description of how these are classified (e.g. immunophenotype). Please provide a table with cell type definitions or flow plots so the readers can interpret these results.

Figure 5: Although conceptually MDS is attractive given expansion of progenitors and impaired differentiation but it is a big leap in the absence of somatic mutations/transcriptomic evidence that point to the SAGA complex in some way. The message needs to be toned down.

- have you interrogated transcriptomic datasets (human MDS CD34⁺ vs healthy CD34⁺ or HR-MDS vs LR-MDS) for differences in SAGA complex gene expression?
- MDS-L cells are an IL-3 dependent cell line generated from an MDS patient. I would not refer to these as MDS cancer cells/mouse model of human MDS. Replace all labels to clearly state that these are MDS-L cells in main and supplemental figures. The data are data is consistent with expansion of KO MDS-L cells, but these cells don't circulate/generate mature cells in vivo. Did the authors perform flow cytometry on retrieved BM-MDSL cells to interrogate their differentiation potential? What do you mean by upregulation of AML/MDS genes (Fig 4S1)?

Minor-

- Line 122. S2e-f is incorrect (S2d and f).
- Figure 1h graph heading is covering the h.
- Figure 3a legend says two guides were used (identified with grey and black dots- only grey dots present). Also says n=6, however plots show n=4 for control, n=7 for Tada2B and n=5 for Taf5l, please clarify.
- Line 616 Figure 3d –Says IFN genes are highlighted in red, they are not. Please fix.
- Line 619 Figure 3d- please fix labels to: *Ifi44* (f), *Oasl2* (g) and *Ddx60* (h).
- Line 620 Figure 3d "E." not sure what this is referring to, is it a mistake?
- Line 704 Figure S3 (g-h) fix label: please change to: Sca1 MFI (g) and MHC-1 MFI (h).
- Figure S3 (g-h) plots say ns. Is this correct? Looks like it should be significant. This is also discussed in text lines 157 and 158- implying this was significant observation.
- Lines 174-175 and Figure 4a (lines 631-632) Both discuss changes in Sca1 MFI, however plot y-axis says "pHSCs frequency normalized to DSMO"?

Version 1:

Reviewer comments:

Reviewer #1

(Remarks to the Author)

I have carefully reviewed the revised manuscript and the responses. The authors have appropriately addressed the major concerns that I had and have also done a good job of stressing the limitations of the screens that were performed. I believe the work is now appropriate for publication.

Reviewer #3

(Remarks to the Author)

I appreciate the authors' thoughtful response to my critiques. They have addressed all my concerns. I congratulate them on this interesting work.

Reviewer #4

(Remarks to the Author)

I commend the authors for the efforts taken to address comments raised at review. I am satisfied that these have been adequately addressed.

Author responses to Reviewer Comments on NCOMMS-25-24960-T

We would like to thank all four Reviewers for their positive and constructive feedback. Our responses to each comment are detailed in **blue font** below.

Reviewer #1 (Remarks to the Author):

In this manuscript, Haney and colleagues present a genome-wide in vivo CRISPR-Cas9 knockout screen in mouse “hematopoietic stem cells”, which identifies structural components of the SAGA complex, particularly Tada2b and Taf5l, as regulators of hematopoiesis. The authors showed ex vivo and in vivo that knockout of these genes results in a block in differentiation from early stem/progenitor cells. Molecular profiling reveals upregulation of interferon-stimulated genes following knockout in the hematopoietic stem/progenitor cell compartment, and loss of these genes enhances outgrowth in a myelodysplastic syndrome model.

The approach presented in this study could be enormously enabling for the field of hematopoietic research, given that the scale of screens that can be conducted using the ex vivo methods described here, could enable many further screens. Moreover, the phenotypes observed are of potential interest. However, there are some limitations to the work as presented. Specifically, the hits from the screen appear insufficiently validated, there is ambiguous interpretation of the phenotypes, and a lack of significant mechanistic insight based on these findings. I detail these specific concerns below.

Major Concerns:

1. The robustness of the screen is unclear. HSPCs were transduced with the library with limited coverage and subjected to extensive ex vivo expansion prior to transplantation, during which functional changes by knockout of critical factors could introduce significant bias. Although controlled subsequently, this likely impacts the power of this screen. Positive controls (e.g. known stem cell regulators) should be evaluated and discussed to confirm the validity of the screen. The authors also compared HSPCs versus individual lineages, are the well-established TFs for lineage commitment enriched in the screen?

We thank the Reviewer for this comment. We have added the following data to address concerns regarding screen robustness and interpretation:

1. To address concerns of library coverage during ex vivo expansion: **new Figure S1b** indicates minimal sgRNA library loss compared to plasmid library following transduction (day 0) and pre-transplantation at day 7.
2. A positive control from the screen, *Runx1*, has now been experimentally validated and shows concordance with screen results (**new Figure S1m-q, new text lines 106-113**).
3. Individual lineages are now compared (**new Figure S1j-l**) highlighting established lineage commitment genes (**new text lines 102-104**).

These newly included data suggest that these screens can identify well established regulators of hematopoiesis and lineage commitment as well as novel regulators such as the SAGA complex. We acknowledge the limitations of this screening approach that the reviewer has pointed out, in a new Limitations of Study section of the Discussion (**new text lines 329-340**).

2. The term “HSCs” is inconsistently applied to both ex vivo cultured HSPCs and phenotypic HSCs. More broadly, the manuscript is ambiguous about the specific stages of differentiation impacted by knockout of the targets in validation experiments. This needs to be more clearly discussed and delineated throughout.

We completely agree with the Reviewer about this terminology. We have updated the manuscript throughout to specify HSCs or HSPCs correctly. We have also highlighted that our cultures are a heterogeneous mix of HSPCs within the introduction (**new text line 85**).

3. The SAGA complex contains multiple modules. Although prior studies on other subunits are

briefly summarized in the introduction, it is perplexing to see that the other components, particularly the catalytic subunits, were not included in the focused CRISPR screen or experimentally explored. Without addressing these, claims about SAGA complex function remain overstated.

We thank the Reviewer for this comment. The Reviewer is correct that not all SAGA subunits were hits in the screen. To more easily understand this, we have added a new graph of the screen results for each SAGA subunits and also included a PyMol 3D heatmap representation (in new **Figures S2a-b**, and copied below).

We believe that there are a number of reasons why specific subunits of the SAGA complex did not appear as strong hits:

1. Some SAGA complex subunits have functional redundancy (e.g. histone acetyltransferase activity can be mediated by KAT2A or KAT2B) so that a single gene KO of one will not result in a phenotype. Given the redundancy of KAT2A/B, we have performed new experiments using a KAT2A/B PROTAC degrader (GSK-699), and could recapitulate many of the phenotypes observed in our *Tada2b/Taf5l* KO experiments. Additionally, we could confirm a global reduction in H3K9ac (deposited by KAT2A/B) and a redistribution of H2Bub (the target of SAGA's USP22 deubiquitinase). Together, these new results provide strong evidence that loss of *Tada2b* or *Taf5l* lead to dysregulation of the SAGA complex (**new Figure 5, new text lines 227-269**).
2. Certain SAGA complex subunits are more important for structural integrity. Interestingly, recent bioRxiv pre-print by Batty et al. 2025 comprehensively investigated how SAGA complex subunits regulate SAGA histone acetyltransferase activity. In close correlation with our screen, they experimentally observed that “*the non-enzymatic SAGA CORE module subunits—TADA1, TAF5L, and TAF6L— as necessary for KAT2A stability, with loss of these subunits disrupting the integrity of SAGA*”, whereas “*Targeting ATXN7, ATXN7L3, USP22 and ENY2, components of the DUB module, did not significantly alter KAT2A levels, consistent with the observation that the DUB module is loosely bound to SAGA and hence dispensable for its structural integrity (Herbst et al, 2021). However, knockout of TADA2B*

and TADA3, components of the HAT module that are in close proximity to KAT2A within SAGA, resulted in a significant drop in KAT2A abundance, indicating that KAT2A protein levels are reduced upon loss of its partners in the HAT module”.

We have added these points to the revised manuscript (**new text lines 241-242**) and added this pre-print to our references (**reference 33**).

4. Related to the previous point, given the well-established role of the SAGA complex in transcriptional regulation, the impact of knockouts on chromatin biology and gene regulation relevant to hematopoiesis should be explored. At a minimum, given the extensive literature in this field, how gene expression changes observed in this setting relate to other perturbations involved in gene regulation that impact HSCs should be examined. Some of this can be achieved by examining publicly available datasets.

We thank the Reviewer for raising this idea. To better understand how the SAGA complex regulates chromatin biology and gene regulation, we have performed new experiments and analyses:

1. We have quantified two histone marks regulated by SAGA, H3K9ac and H2Bub, by intracellular flow cytometry (**new Figure 5c-d, S5c-d**). This analysis identified that loss of *Tada2b* (or *Taf5l*, *Tada1*) led to a 20-30% reduction in H3K9ac levels, but no detectable difference in H2Bub levels.
2. We performed ChIPmentation to assess H3K9ac and H2Bub genomic distribution within *Tada2b*-KO and control CD201⁺CD150⁺KSL HSCs (**new Figures 5h, S5g-k**). This identified significant changes in the distribution of H2Bub following *Tada2b* loss. Interestingly, H3K9ac was not redistributed by loss of *Tada2b* but enrichment was reduced.
3. To further investigate chromatin biology, we performed ATAC-seq on *Tada2b*-KO and control CD201⁺CD150⁺KSL HSCs (**new Figure 5e-g**). In line with a recent ATAC-seq screen that identified SAGA complex members as regulators of global chromatin accessibility, we identified an overall reduction in the number of chromatin accessibility sites.
4. Given the global reduction in H3K9ac levels (and recent Batty et al. bioRxiv 2025 pre-print), we hypothesized that loss of SAGA histone acetyltransferase activity was a major dysregulated pathway in *Tada2b*-KO HSCs. We therefore assessed the consequences of a KAT2A/B inhibitor/degrader (GSK-699) on HSPCs, and could recapitulate several *Tada2b*-KO phenotypes including:
 - a. Reduction in H3K9ac levels (**new Figure 5k**)
 - b. Increased immunophenotypic HSC and MkP cells in HSPCs (**new Figure 5n-o**)
 - c. Increased Sca-1 expression within cultured HSPCs (**new Figure 5m**)
 - d. Strong correlation in gene expression changes by RNA-seq (**new Figure 5p-q**)

We hope that the Reviewer agrees with us that these new results provide important new insights into how SAGA complex members regulate HSPC activity.

5. (sc)RNA-seq results showed upregulation of IFN genes, but the causal relationship between this signature and the observed phenotype as well as the unchanged key markers in vivo remain largely underdeveloped. This requires further development, as discussed in point 4 above.

We appreciate this Reviewer comment. To investigate the causal relationship between IFN and *Tada2b*, we performed a new transplantation assay comparing *Ifnar1*-KO *Tada2b*-KO HSPCs with *Ifnar1*-KO *Rosa26*-KO HSPCs (**Figure S4i**). *Ifnar1*-KO did not reverse the *Tada2b*-KO differentiation block phenotype seen in our previous *Ifnar1*-competent HSPC transplantation assays (in **Figure 2**). These results suggest that loss of IFNAR1 signalling alone was insufficient to rescue the in vivo differentiation defect. We have updated the text accordingly (**new text lines 223-224**).

6. The use of an MDS cell line appears a little arbitrary as there is no evidence presented that the candidate genes are mutated, dysregulated, or otherwise implicated in MDS. Can better rationale for this exploration be presented?

We thank the Reviewer for their thoughtful critique regarding the rationale for using the MDS-L cell line model in our study. Our use of the MDS-L model was motivated by convergent phenotypes between SAGA complex loss-of-function and hallmark features of MDS, including (1) the blocked hematopoietic differentiation and progenitor cell accumulation phenotypes, and (2) interferon (IFN) pathway activation. We have updated the text to provide a more detailed justification (**new text lines 273-274**).

To provide readers with more context, we have also added two new analyses:

1. We re-analyzed public transcriptomic datasets from MDS patient HSPCs and confirmed an overlap in upregulated ISGs identified in CD34⁺ cells from MDS patients. This data is presented in **Figure S6a** and copied below.

2. We queried the COSMIC (v97) database to assess the frequency of somatic mutations in *TADA2B*, *TAF5L*, *TAF8*, *TAF13*, *USP22*, *KAT2A*, and *KAT2B* across MDS, AML, and CMML patient cohorts. This analysis is summarised in **Figure S6f** and copied below, which found that these genes are mutated at low frequencies (0.1–1.2%) across these disease contexts.

Minor Concerns:

1. Figures need revision-1c, 3d, and 4f lack proper labeling, 1e uses incorrect nomenclature for mouse

genes, 2g used inconsistent color schemes.

We thank the Reviewer for catching these figure issues and have corrected them.

Reviewer #2 (Remarks to the Author):

We thank Reviewer #2 for their contribution to the peer-review process.

Reviewer #3 (Remarks to the Author):

In this manuscript Haney et al describe an *in vivo* CRISPR/Cas9 screen to evaluate genes required for normal hematopoietic differentiation. They develop an *ex vivo* culture system to maintain HSCs allowing expansion and lentiviral transduction, followed by transplantation into lethally irradiated recipient mice and an *in vivo* readout. Using this elegant system they identify components of the SAGA complex as mediators of hematopoietic differentiation. They show that loss of these components leads to upregulation of the interferon response mouse hematopoietic cells. Finally, they show that knockout of these genes in the MDS-L model promotes engraftment and growth of these cells in a xenograft model. The *in vivo* screening system is exciting, and the role of SAGA in hematopoiesis is intriguing. Overall, the paper is clearly written, although lacking somewhat in discussion, but there is a lack of clarity as to what SAGA's role in hematopoietic differentiation is. I lay out these concerns plus some additional points below:

-The details of the whole-genome screen are not clearly described. Was each sub-pool of the library used to infect 50 million HSCs and then transplanted into 65 recipients? Or is the 50 million cells and 65 mice the total for the entire screen (all the sub pools combined)? Was the screen done in singlicate or replicate? Having specific numbers and a clearer description is helpful in assessing the representation and potential coverage of the screen. The schematic in Figure 1a also does not contain much detail or add much in explaining how the screen was performed. One of the more novel and exciting aspects of this report is the whole-genome *in vivo* screening approach, so additional details are important.

We thank the Reviewer for pointing out that the screening methodology was not sufficiently clear in the methods section. As detailed in the revised Methods section (**lines 371-387**), each sub-library (containing ~20,000 elements; total of 10 sub-libraries for the genome-wide screen) was infected into 50 million HSPCs (at an MOI of ~0.3 to achieve 20-30% transduction), to give a coverage of 500-750x *in vitro*. Following puromycin selection and recovery, >10 million HSPCs were injected into each mouse cohort (1-2 million per mouse) in an attempt to maintain this coverage *in vivo*. The genome-wide screen and the gene expression screen were performed in singlet and the top hits of interest (SAGA complex) were extensively validated. We want to emphasize that while we believe this to be a powerful technique, the focus of this manuscript is on the discovery of the role of the SAGA complex in hematopoiesis and not this screening approach as a resource or novel method.

-1-2 million HSCs were transplanted per mouse, however generally it is thought that many fewer HSC's actually contribute to adult hematopoiesis (maybe around 100,000), so it's unclear how this would impact representation within the screen. Did the authors evaluate sgRNA representation in the pooled HSC's prior to transplantation, or in either the HSC or mature cell pools? If possible, some measure of this should be included in the supplemental methods/data.

We thank the Reviewer for their concern. we now have included screen representation prior to transplantation and in the mature cell pools (**Figure S1b-c**). The sgRNA libraries are well represented *ex vivo*, but this does drop *in vivo*, with ~50% of sgRNAs detected. It is worth noting that this still represents ~5 sgRNAs per gene, although there is clearly a bottleneck *in vivo*.

We explicitly acknowledge this loss of library coverage *in vivo* as a limitation of this study in the revised Discussion (**new text lines 329-340**). With sgRNA library used in this study (and numbers of mice transplanted), we believe perfect representation *in vivo* is unlikely to be feasible. We believe that newer barcoding sgRNA screening approaches should help improve *in vivo* screens, such as that published by Loket et al. (Blood 2025) during these revisions. This is also discussed in our revised manuscript.

-“Here we refer to the cells in these cultures as HSCs; we acknowledge that these cultures include hematopoietic progenitor cells but for simplicity we will refer to these as HSC cultures.” Why not refer to these as hematopoietic stem and progenitor cells (HSPCs), which would be more accurate, and the abbreviation is just as easy to use throughout the paper? I feel that this term is more accurate and will avoid potential confusion.

We completely agree with the Reviewer about this terminology. We have updated the manuscript throughout to specify HSCs or HSPCs correctly. We have also highlighted that our cultures are a heterogeneous mix of HSPCs within the introduction (**new text line 85**).

-Were these the only 4 members of the SAGA complex to score in the screens? What about other members of the complex or the catalytic subunit, KAT2A?

We thank the Reviewer for this comment. The Reviewer is correct that not all SAGA subunits were hits in the screen. To more easily understand this, we have added a new graph of the screen results for each SAGA subunits and also included a PyMol 3D heatmap representation (in new **Figures S2a-b**, and copied below).

We believe that there are a number of reasons why specific subunits of the SAGA complex did not appear as strong hits:

1. Some SAGA complex subunits have functional redundancy (e.g. the histone acetyltransferases *Kat2a/Kat2b*) so that a single gene KO of one will not result in a

phenotype. However, to investigate the role of KAT2A/B, we have performed new experiments using a KAT2A/B PROTAC degrader, and can recapitulate many of the phenotypes observed in our *Tada2b/Taf5l* KO experiments. Additionally, we could confirm a global reduction in H3K9ac (deposited by KAT2A/B) and a redistribution of H2Bub (the target of SAGA's USP22 deubiquitinase). Together, these new results provide strong evidence that loss of *Tada2b* or *Taf5l* lead to dysregulation of the SAGA complex (**new Figure 5, new text lines 227-269**).

2. Certain SAGA complex subunit are more important for structural integrity. Interestingly, recent bioRxiv pre-print by Batty et al. 2025 comprehensively investigated how SAGA complex subunits regulate SAGA histone acetyltransferase activity. In close correlation with our screen, they experimentally observed that “*the non-enzymatic SAGA CORE module subunits—TADA1, TAF5L, and TAF6L—as necessary for KAT2A stability, with loss of these subunits disrupting the integrity of SAGA*”, whereas “*Targeting ATXN7, ATXN7L3, USP22 and ENY2, components of the DUB module, did not significantly alter KAT2A levels, consistent with the observation that the DUB module is loosely bound to SAGA and hence dispensable for its structural integrity (Herbst et al, 2021). However, knockout of TADA2B and TADA3, components of the HAT module that are in close proximity to KAT2A within SAGA, resulted in a significant drop in KAT2A abundance, indicating that KAT2A protein levels are reduced upon loss of its partners in the HAT module*”.

We have added these points to the revised manuscript (**new text lines 241-242**) and added this pre-print to our references (**reference 33**).

-It's a little unclear what is being proposed as a model for the function of SAGA in hematopoiesis. There are potentially multiple different SAGA complexes or modules with different component members. Is the hypothesis that the components of SAGA complex identified in this screen make up a particular complex important for directly regulating a subset of genes important for hematopoiesis? Are there published ChIPseq data for KAT2A or other SAGA complex members that might shed light on which of these effects is direct?

We appreciate this comment, which stimulated us to further investigate the mechanisms underlying this interesting phenotype with new experiments.

First, we were stimulated by a recent pre-print from Batty et al., (bioRxiv 2025) who have found that TADA2B, TAF5L, and TADA1 disrupt KAT2A histone acetyl-transferase (HAT) activity and stability in HAP1 cells (a human cell line). We therefore quantified H3K9ac levels in control and KO HSPCs (**new Figure 5c, S5c**). We observed a significant loss of H3K9ac levels in this setting, in line with disruption of SAGA HAT activity. To assess the effect on the SAGA deubiquitinase (USP22) activity, we also quantified H2Bub levels, however, no change in H2Bub were observed (**new Figure 5d, S5d**). We further assessed H3K9ac and H2Bub genomic distribution within *Tada2b*-KO and control HSCs (**new Figure 5h, S5g-i**). This identified a significant redistribution of H2Bub following *Tada2b* loss. Interestingly, H3K9ac was not redistributed by loss of *Tada2b* but enrichment was reduced (**Figure S5j-k**). Consistent with a global reduction in H3K9ac (a modification associated with open chromatin), we also observed an overall reduction in open chromatin regions in *Tada2b*-KO HSCs by ATAC-seq (**Figure 5e**). We therefore conclude that loss of *Tada2b* dysregulates both H3K9ac and H2Bub chromatin marks, and reduces chromatin accessibility.

Second, we were stimulated by a recent paper by Nguyen et al (Science Advances 2024) who suggested that loss of *Kat2a/Kat2b* in intestinal stem cells led to increased interferon activation and signalling via disruption of mitochondrial integrity and activity. We therefore assessed mitochondrial activity within our control and *Tada2b*-KO HSPC cultures. We observed a significant decrease in mitochondrial activity without altering overall mitochondrial cell mass (**new Figure 5a, S5a-b**). Corresponding with this partial loss in mitochondrial activity, *Tada2b*-KO HSPCs displayed greater sensitivity to low doses of a mitochondrial complex I inhibitor IACS-010759 (**new Figure 5b**). We

therefore conclude that SAGA complex KOs result in reduced mitochondrial activity. We speculate that this is a driver of IFN secretion and downstream IFN signalling in the *Tada2b*-KO setting.

Please note that we did also check for relevant KAT2A/SAGA ChIP-seq data but were unable to find any in an appropriate cell type to be directly relevant for this study.

-The focus here is on SAGA-mediated regulation of the interferon response. However, it is not clear that this is mediating the hematopoietic phenotype. Is there a hematopoietic phenotype to the *Ifnar1* knockout? Did any component of the interferon response score in the original genome-wide screen? What is mediating the hematopoietic differentiation defect?

We thank the Reviewer for stimulating us to further interrogate the interferon response in this phenotype. As requested, we also rechecked our genome-wide screen for interferon response genes but were unable to identify any as hits. We performed transplantation studies using *Rosa26*-KO *Ifnar1*-KO or *Tada2b*-KO *Ifnar1*-KO HSPCs. Interestingly, the *Ifnar1*-KO did not rescue the *Tada2b*-KO phenotype of reduced PB output and increased Lin- bone marrow. We have added this new data in **Figure S4i** and updated the text to include this new insight (**new text lines 223-224**).

As detailed above, we have found that SAGA complex KOs disturb (1) H3K9ac and H2Bub in HSPCs, (2) chromatin accessibility, and (3) mitochondrial activity. It is likely that all contribute to the haematopoietic differentiation defect but are difficult to disentangle experimentally. We have updated the Discussion to speculate on this mechanism (**new text lines 298-313**).

-What is the SAGA target that mediates the hematopoietic differentiation defect? Can the authors cross-reference the list of genes or pathways up- or down-regulated in the SAGA mutant cells with the genes that scored in the genome-wide screen? This might provide insight into the mechanistic role that SAGA plays in hematopoiesis.

We thank the Reviewer for the suggestion to overlap our RNA-seq DEGs with our CRISPR screen hits to identify mechanistic roles. Unfortunately, this analysis did not identify any overlapping genes. This may well be due to the differentiation defect occurring in downstream progenitor cells while our RNA-seq comes from primitive HSCs.

As detailed above, we now have evidence that loss of SAGA complex members results in dysregulation of H3K9ac (**new Figures 5c, S5c**), H2Bub (**new Figure 5d, S5d**), chromatin accessibility (**new Figure 5e**), interferon signalling (**Figures 3-4**), and mitochondrial activity (**new Figure 5a-b**). We believe that the differentiation defect is likely to be driven by the sum of these molecular perturbations and will be complex to tease apart. We have updated the discussion to highlight this (**new text lines 298-313**).

-The authors make the case that knockout of SAGA complex members supports the development of MDS/AML. To my knowledge, mutations in SAGA complex members have not been reported in clinical cohorts of MDS/AML. Are there any data to suggest that there is up- or down-regulation of these genes during disease formation, or anything else to suggest that changes in SAGA function participate in disease biology in humans?

We thank the Reviewer for this comment. Our use of the MDS-L model was motivated by convergent phenotypes between SAGA complex loss-of-function and hallmark features of MDS, including (1) the blocked hematopoietic differentiation and progenitor cell accumulation phenotypes, and (2) interferon (IFN) pathway activation. We have updated the text to provide a more detailed justification (**text lines 272-273**).

To provide readers with more context, we have also added two new analyses:

1. We re-analyzed public transcriptomic datasets from MDS patient HSPCs and confirmed an overlap in upregulated ISGs identified in CD34⁺ cells from MDS patients. This data is presented in **Figure S6a** and copied below.

2. We queried the COSMIC (v97) database to assess the frequency of somatic mutations in *TADA2B*, *TAF5L*, *TAF8*, *TAF13*, *USP22*, *KAT2A*, and *KAT2B* across MDS, AML, and CMML patient cohorts. This analysis is summarised in **Figure S6f** and copied below, which found that these genes are mutated at low frequencies (0.1–1.2%) across these disease contexts.

Minor issues:

-Page 2, Line 65-67 (and throughout the manuscript): Mouse genes should have the first letter capitalized, and the remainder should be lower case letters and then italicized. Mouse protein names should have all their letters capitalized. So, “Kat2a” should be KAT2A.

See https://www.informatics.jax.org/mgihome/nomen/short_gene.shtml for further reference.

We thank the Reviewer for catching this error and have corrected the manuscript text here (and throughout) accordingly.

-Figure 1e-h: The gene names are capitalized here, but should be lower case.

We thank the Reviewer for catching this error and have corrected the figure (and other figures) accordingly.

-Figure 3d: the legend says, “IFN response genes are highlighted in red.” However, no genes are highlighted in red in the figure.

We apologise for this text error; the figure legend has been corrected to read “IFN response genes are underlined” (text line 864).

-Figure 4a: The figure suggests that there is a highly significant statistical difference between the “Control +TBKi” and “Tada2b KO +TBKi” conditions. However, by eye these two conditions look identical. Can the authors confirm whether this is correct or not?

We thank the Reviewer for catching this and have corrected it. To confirm those two conditions are not statistically significant.

-Methods describing analysis of bulk RNAseq are not included in the methods section.

We thank the Reviewer for noting this omission and have updated the Methods section to include details of our bulk RNA-seq analysis (new text lines 510-516).

Reviewer #4 (Remarks to the Author):

Haney et al performed a genome-wide CRISPR KO screen in ex vivo expanded murine cKit+ cells and following transplantation identified gene edits corresponding to members of the SAGA complex that were enriched in HSPCs vs mature cells in vivo. The results were confirmed using a focused CRISPR screen targeting around 2000 genes. The authors focus on Tada2b and Taf51 KO and show an accumulation of ST-HSC/progenitors in bone marrow (but not LT-HSCs). Leveraging single cell transcriptomics in CD201+CD150+KSL (ex vivo cultures), the authors show activation of IFN genes in KO cells compared to controls. Over expression of Tada2b dampened these signatures. Increased IFN secretion from cultured KO Sca1+ cells, reduced numbers of Sca1+ cells following GSK-8612 and Tada2b edits in ifnar KO cells showed elevation of alternate sets of genes (that maintained elevation of HSCs) but not of IFN genes. Gene edits were then created in MDS-L cells and accumulation of these in xenograft BM/Spleen and activation of IFN gene sets in harvested cells. This is an elegant manuscript with well-designed experiments and clear writing. The mouse experiments in particular are very well done.

Figure 1: Was this a single expansion culture and transplant of cells divided across 65 recipients with material pooled and sorted to identify sg enriched in stem/progenitors? If so, the schematic could be made clearer. How many recipient mice were used for the 2000 gene (SAGA enriched) screen?

We thank the Reviewer for pointing out that the screening methodology was not sufficiently clear in the methods section. As detailed in the revised Methods section (lines 371-387), each sub-library (containing ~20,000 elements; total of 10 sub-libraries for the genome-wide screen) was infected into 50 million HSPCs (at an MOI of ~0.3 to achieve 20-30% transduction), to give a coverage of 500-750x in vitro. Following puromycin selection and recovery, >10 million were injected into each mouse cohort (1-2 million per mouse) in an attempt to maintain this coverage in vivo. For the genome-wide screen, a total of 65 mice were transplanted. For the gene expression screen, a total of 24 mice were transplanted. The genome-wide screen and the gene expression screen were performed in singlet and the top hits of interest (SAGA complex) were extensively validated. We want to emphasize that while we believe this to be a powerful technique, the focus of this manuscript is on the discovery of the role of the SAGA complex in hematopoiesis and not this screening approach as a resource or novel method.

As requested, we have updated the screen schematic to highlight additional details (Figure 1a).

Figure 2: Please show the flow gating strategy used for immunophenotyping. Particularly for the BM LT-HSCs vs ST/LT-HSCs and pHSC/MPPs (Figure 2 c-f) given the difference in donor chimerism (LT-HSCs being similar PB WBCs).

We apologise for not including representative flow cytometry gating in our initial submission. These have been added to **Figure S7**.

Figure 3:

- Phenotypic KO LT-HSCs expanded ex vivo (Fig 3a/128-130) but KO LT-HSCs were lower in vivo? Showing total cell expansion numbers rather than frequencies and flow cytometry gating will be helpful (ditto 3i and S4g)

We would like to clarify that the ex vivo immunophenotypic HSC definition CD201⁺CD150⁺KSL is not as functionally pure as the in vivo LT-HSC population. Serial engraftment capacity has been shown to be restricted to this cell fraction but only ~1:3 CD201⁺CD150⁺KSL from culture engraft in single cell transplantation assays (at a 12-week endpoint) according to Che et al., EMBO Reports 2022. Our best guess is this ex vivo HSC population is a mixture of LT-HSCs, ST-HSCs and MPPs. We would therefore not like to directly compare in vivo and ex vivo HSC frequencies, given the differences in functional heterogeneity. However, the expansion of this population ex vivo could be driven by increases in downstream cell types, which are also expanded in vivo.

We appreciate the suggestion of presenting cell numbers rather than frequencies and representative flow cytometry gating. These have been added to **Figures S3a** and **S4h**, and representative gating is now included in **Figure S7c**.

- Comparisons were limited to HSCs in G1- was the IFN signature also increased in MPPs in G1?

We thank the Reviewer for this comment. We do see IFN upregulation in all clusters of our scRNA-seq analysis. This was already included in **Tables S3-4**, but not discussed in the manuscript. We have now noted this insight in the main text (**text lines 189-190**).

- MkP numbers appear to be increased in Tada2b and Taf5l KO?

We thank the Reviewer for this comment, which stimulated us to check for MkP frequencies within our HSPC cultures. As predicted by the scRNA-seq experiment, we did observe increases in this cell population (CD41⁺CD150⁺c-Kit⁺Lineage⁻), which are now included in a new **Figure S3h**.

We could also confirm that the same phenotype was observed following treatment with the KAT2A/B inhibitor/degrader GSK-699, implicating that this megakaryocyte bias was linked to the loss of SAGA HAT stability following loss of *Tada2b* or *Taf5l*. This new data has been added to **Figure 5o**.

Overall, there is a lot of cell type nomenclature throughout this paper (MPP/HSC, pHSC, LTHSC, STHSC, Mkp etc) with very little description of how these are classified (e.g. immunophenotype). Please provide a table with cell type definitions or flow plots so the readers can interpret these results.

We appreciate the suggestion and have included a new Table to include our immunophenotypic definition of these populations (**text line 449**). Representative flow cytometry gating plots are also now available in **Figure S7**.

Figure 5: Although conceptually MDS is attractive given expansion of progenitors and impaired differentiation but it is a big leap in the absence of somatic mutations/transcriptomic evidence that point to the SAGA complex in some way. The message needs to be toned down.

We thank the Reviewer for the comment. As requested, we have toned down this message in the abstract and main text. Our use of the MDS-L model was motivated by convergent phenotypes between SAGA complex loss-of-function and hallmark features of MDS, including (1) the blocked hematopoietic differentiation and progenitor cell accumulation phenotypes, and (2) interferon (IFN) pathway activation. We have updated the text to provide a more detailed justification (**new text lines 272-273**).

To provide readers with more context, we have also added two new analyses:

1. We re-analyzed public transcriptomic datasets from MDS patient HSPCs and confirmed an overlap in upregulated ISGs identified in CD34⁺ cells from MDS patients. This data is presented in **Figure S6a** and copied below.

2. We queried the COSMIC (v97) database to assess the frequency of somatic mutations in *TADA2B*, *TAF5L*, *TAF8*, *TAF13*, *USP22*, *KAT2A*, and *KAT2B* across MDS, AML, and CMML patient cohorts. This analysis is summarised in **Figure S6f** and copied below, which found that these genes are mutated at low frequencies (0.1–1.2%) across these disease contexts.

- have you interrogated transcriptomic datasets (human MDS CD34⁺ vs healthy CD34⁺ or HR-MDS vs LR-MDS) for differences in SAGA complex gene expression?

We thank the Reviewer for this idea. However, we could not identify differential expression of SAGA complex members in human MDS transcriptomic datasets. By contrast, the same IFN genes we see upregulated in SAGA complex member KO HSCs have long been documented as upregulated in human MDS (e.g. Pellagatti et al 2006. Blood). This has also been reported more recently with single-cell RNA-seq data sets of MDS patients (Guo et al., Journal of Translational Medicine 2024).

- MDS-L cells are an IL-3 dependent cell line generated from an MDS patient. I would not refer to these as MDS cancer cells/ mouse model of human MDS. Replace all labels to clearly state that these are MDS-L cells in main and supplemental figures. The data are data is consistent with expansion of

KO MDS-L cells, but these cells don't circulate/generate mature cells in vivo. Did the authors perform flow cytometry on retrieved BM-MDSL cells to interrogate their differentiation potential? What do you mean by upregulation of AML/MDS genes (Fig 4SI)?

We appreciate the advice and have updated the text and figures to specify these are MDS-L cells.

Regarding MDS-L differentiation potential, we went back to frozen bone marrow samples from our MDS-L xenotransplantation experiment and checked several human lineage markers by flow cytometry: CD34 (as a stem/progenitor cell marker), CD41 (as a megakaryocyte marker) and CD33 (as a myeloid marker). These were selected because MDS-L cells have been reported to have megakaryocyte and myeloid differentiation potential. We observed high expression of all markers in control and KO samples, with no differences identified. Given this negative data does not add materially to the manuscript, we have not included it in the revised manuscript.

We apologise that we were not clear regarding the upregulation of AML/MDS genes. We have revised this sentence of the text (**text line 286**).

Minor-

- Line 122. S2e-f is incorrect (S2d and f).

We thank the Reviewer for catching this and have corrected it (**text line 152**).

- Figure 1h graph heading is covering the h.

We thank the Reviewer for catching this and have corrected Figure 1h.

- Figure 3a legend says two guides were used (identified with grey and black dots- only grey dots present). Also says n=6, however plots show n=4 for control, n=7 for Tada2B and n=5 for Taf5I, please clarify.

We apologise for this discrepancy. We have corrected **Figure 3a** with the correctly coloured dots, and also updated the figure legend with the correct replicate numbers (**text line 850**).

- Line 616 Figure 3d –Says IFN genes are highlighted in red, they are not. Please fix.

We apologise for this text error; the figure legend has been corrected to read “IFN response genes are underlined” (**text line 864**).

- Line 619 Figure 3d- please fix labels to: Ifi44 (f), Oasl2 (g) and Ddx60 (h).

We thank the Reviewer for catching this and have corrected it (**text line 867**).

- Line 620 Figure 3d “E.” not sure what this is referring to, is it a mistake?

We apologise for this text error. This has been deleted.

- Line 704 Figure S3 (g-h) fix label: please change to: Sca1 MF1 (g) and MHC-1 MFI (h).

We thank the Reviewer for catching this and have corrected it (**Figure S3i-j**).

- Figure S3 (g-h) plots say ns. Is this correct? Looks like it should be significant. This is also discussed in text lines 157 and 158- implying this was significant observation.

We thank the Reviewer for catching this. We rechecked these statistics and the differences are non-significant. We have updated the main text to avoid confusion (**text line 194**).

- Lines 174-175 and Figure 4a (lines 631-632) Both discuss changes in Scal MFI, however plot y-axis says “pHSCs frequency normalized to DSMO”?

We thank the Reviewer for catching this error and have corrected this plot label to “Fold change in Scal MFI”.